# Tidal dynamics and mangrove carbon sequestration during the Oligo–Miocene in the South China Sea

Daniel S. Collins[1], Alexandros Avdis[1], Peter A. Allison[1], Howard D. Johnson[1], Jon Hill[2], Matthew D. Piggott[1], Meor H. Amir Hassan[3] & Abdul Razak Damit[4]

Modern mangroves are among the most carbon-rich biomes on Earth, but their long-term ($\geq 10^6$ years) impact on the global carbon cycle is unknown. The extent, productivity and preservation of mangroves are controlled by the interplay of tectonics, global sea level and sedimentation, including tide, wave and fluvial processes. The impact of these processes on mangrove-bearing successions in the Oligo–Miocene of the South China Sea (SCS) is evaluated herein. Palaeogeographic reconstructions, palaeotidal modelling and facies analysis suggest that elevated tidal range and bed shear stress optimized mangrove development along tide-influenced tropical coastlines. Preservation of mangrove organic carbon (OC) was promoted by high tectonic subsidence and fluvial sediment supply. Lithospheric storage of OC in peripheral SCS basins potentially exceeded 4,000 Gt (equivalent to 2,000 p.p.m. of atmospheric $CO_2$). These results highlight the crucial impact of tectonic and oceanographic processes on mangrove OC sequestration within the global carbon cycle on geological timescales.

[1] Department of Earth Science and Engineering, Imperial College London, South Kensington Campus, London SW7 2AZ, UK. [2] Environment Department, University of York, Heslington, York YO10 5DD, UK. [3] Geology Department, University of Malaya, Kuala Lumpur 50603, Malaysia. [4] No. 9, Simpang 265-254, Kampong Bukit Bunga, Jerudong BG3322, Brunei Darussalam. Correspondence and requests for materials should be addressed to D.S.C. (email: dscollins.geo@gmail.com) or to P.A.A. (email: p.a.allison@imperial.ac.uk).

                                                                 1

Despite accounting for only c. 0.04% ($0.14 \times 10^6$ km²) of the present-day global ocean area[1], mangroves are responsible for c. 4% (c. $24 \times 10^{12}$ g organic carbon (OC) year$^{-1}$) of mean annual carbon burial in the global ocean (Fig. 1a)[2–6]. High OC burial in mangroves on centennial timescales (c. 170 g OC m$^{-2}$ year$^{-1}$)[3,5] is due to high rates of primary productivity (1,110 g OC m$^{-2}$ year$^{-1}$)[3] and exceptionally efficient sediment trapping by complex roots[7], which promotes rapid sediment accretion (c. 5 mm year$^{-1}$)[3,5]. Consequently, mangroves are a major oceanic 'hotspot' for OC burial (Fig. 1b)[6,8,9].

On geological timescales ($\geq 10^6$ years), the preservation potential of OC in coastal–shelf sediments depends on the relative rate and magnitude of sedimentation, subsidence and erosion (Fig. 1c). Sediment supply, accommodation (the space available for sediment accumulation)[10] and depositional process control the rate of sedimentation, sediment thickness and character of sedimentary layering (stratigraphic architecture)[11]. Subsidence is driven by tectonics, sediment loading and compaction. Erosion occurs by several mechanisms with the repeat time between successive erosion events varying across 12 orders of magnitude (Fig. 1c)[11]. The effectiveness of erosional

**Figure 1 | OC burial and preservation in mangrove systems.** (**a**) Absolute yield of OC buried annually in each major oceanic sediment type[3,6]. (**b**) Area-normalized annual OC burial yields[3,6]. Deltaic and non-deltaic continental shelf environments are combined into 'continental shelf' due to poor constraints on deltaic area[6]. (**c**) The range and timescales of sedimentary erosion processes[11] and geological controls on sediment burial and lithospheric preservation in the context of mangrove systems. The shaded triangles indicate the main controls on preservation, including the importance of erosion processes.

processes is generally greater as the frequency between successive erosive events increases (Fig. 1c)[11]. Sediment is preserved on geological timescales when it is buried deeper than the depth of erosion and accumulates faster than the repeat time of each successive erosion event. Consequently, sediment preservation is enhanced by higher long-term rates of accommodation space creation, sediment supply and subsidence[10,11]. On short timescales ($\leq 10^3$ years), the mangrove biome contributes to increased OC preservation by trapping and stabilizing sediment and reducing the magnitude of erosion induced by waves, tides, storms and extreme events (for example, tsunamis)[7,12].

The geographical ($10-10^3$ km) distribution of modern mangroves is controlled by climate (temperature, precipitation and storms), salinity and sea-level fluctuations[12,13]. Mangroves are most extensive along tide-dominated, tropical–subtropical shorelines[12] because: (1) higher tidal range increases the intertidal area for mangrove colonization; (2) tidal inundation excludes colonization by salinity-intolerant flora; and (3) tidal action produces tidal channels and lagoons and generally increases coastline rugosity and protection from waves[7,12,13]. Mangroves also occur within abandoned areas of fluvial-dominated deltas, including inactive channel systems, around protected coastal embayments and lagoons, and along some wave-dominated shorelines[13–15]. At present, around 40% of the world's mangroves are in Southeast Asia[1]. However, this region is geologically

dynamic and has undergone complex and significant plate tectonic and palaeogeographic changes during the Oligo–Miocene[16]. Hence, coastal processes and geomorphology, intertidal vegetation and coastal–shelf OC burial are all likely to have varied during the past 25–35 Myr. Furthermore, this region contains significant hydrocarbon accumulations that were principally sourced by terrestrial- and mangrove-dominated OC preserved in coastal-shelf and deep-water sediments[17,18].

Here we integrate palaeogeographic reconstructions, palaeotidal modelling and sedimentary analysis and suggest that higher tidal range and stronger tidal currents caused widespread mangrove development along ancient shorelines in the Oligo–Miocene South China Sea (SCS). Furthermore, this has decreased through time due to tectonic-driven changes in tidal dynamics. We also show that preservation of mangrove OC was optimized along tide-dominated coastlines in the Oligo–Present SCS, which could be a significant component of the global carbon cycle on geological timescales.

## Results

**Oligo–Miocene palaeogeography of Southeast Asia.** Southeast Asia comprises a complex mosaic of geological terranes assembled along a system of active and extinct subduction and collisional zones. Critically, the region is also located at the triple junction between three major tectonic plates: the Eurasian,

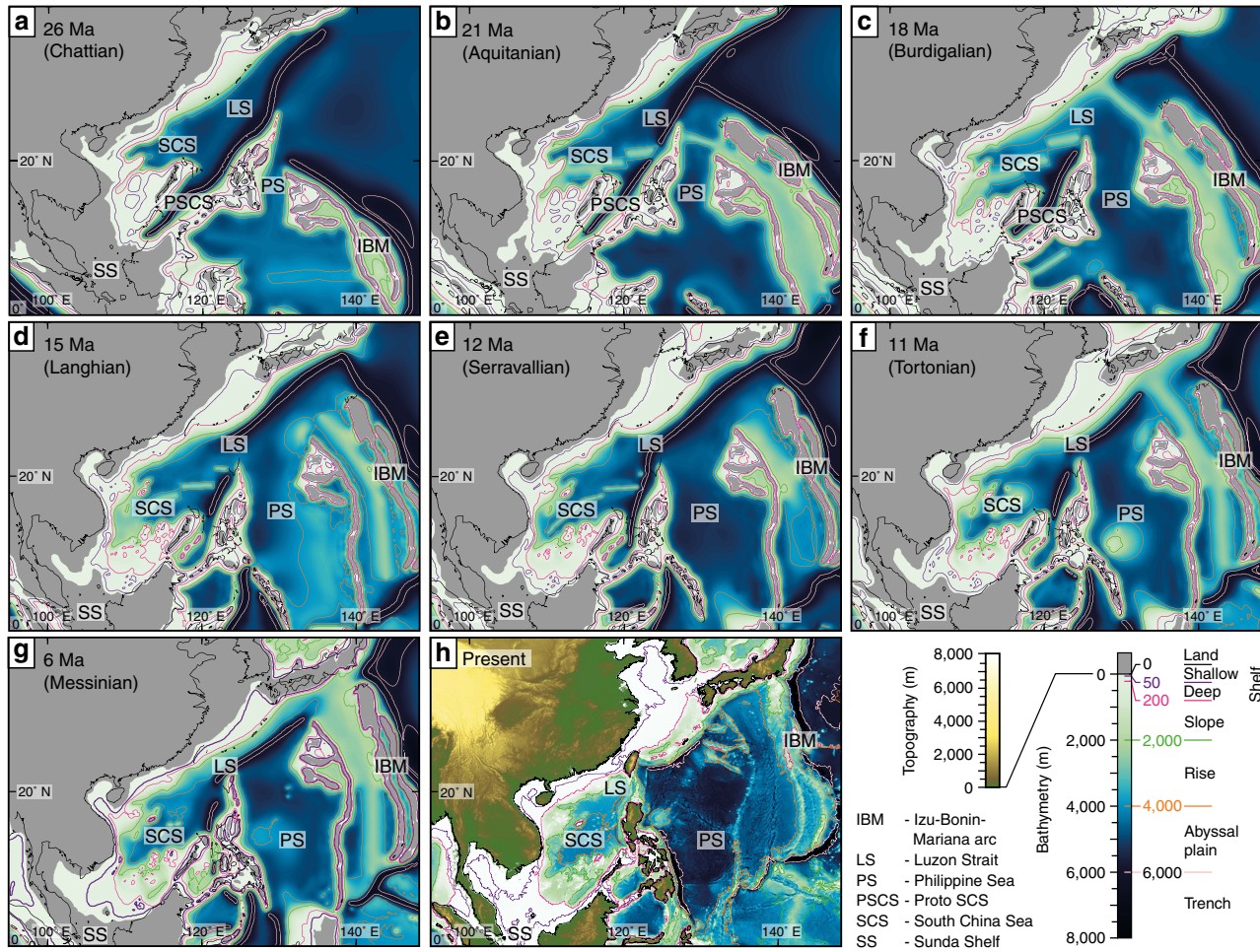

**Figure 2 | Palaeobathymetric reconstructions for the Late Oligocene–Late Miocene Southeast Asia.** These reconstructions are based on sea-level highstand for eight timeslices: (**a**) 26 Ma (Chattian); (**b**) 21 Ma (Aquitanian); (**c**) 18 Ma (Burdigalian); (**d**) 15 Ma (Langhian); (**e**) 12 Ma (Serravallian); (**f**) 11 Ma (Tortonian); (**g**) 6 Ma (Messinian); and (**h**) Present. See Supplementary Fig. 1 for palaeogeographies of **a**–**g**, including sensitivity analyses.

Indo-Australian, and Pacific plates[16]. Palaeogeographic modelling for the Oligo–Miocene in Southeast Asia synthesizes diverse published and unpublished sedimentological, stratigraphic and palaeogeographic data (see Methods section).

The Luzon Strait (LS) between Taiwan and the Philippines was wider in the past than the present day (Fig. 2). The LS decreased in width from c. 1,300 km in the Late Oligocene (Fig. 2a) to c. 500 km in the Late Miocene (Fig. 2g), due to northward movement of the Philippines relative to China and Taiwan. The modern LS is c. 350 km wide (Fig. 2h) and is critical for transferring oceanic flows, tides and waves from the Pacific Ocean into the SCS[19]. Furthermore, the Philippines presently have a significant blocking effect on tropical storms, which move westwards from the Pacific Ocean towards the SCS[20]. Therefore, a wider LS would have allowed more tide, wave and storm-wave energy to propagate into the SCS during the Oligo–Miocene. Furthermore, in the present day, a shallow (<100 m) Sunda Shelf permits throughflow of water from the Pacific, entering the SCS through the LS and exiting through several seaways into the Indian Ocean. However, the Sunda Shelf was emergent throughout the Oligo–Miocene[16] (Fig. 2), which prevented oceanic connection to the west; more tide, wave and storm-wave energy was trapped within the SCS.

Uncertainty in the palaeogeographic position of Palawan stems from a lack of conclusive data regarding Jurassic–Palaeogene plate reconstructions and the genesis of the Northwest Borneo–Palawan Trough[21,22]. As a result, two end-member interpretations of Palawan's position in Oligo–Miocene reconstructions are included as sensitivity studies. In our base-case palaeogaphic interpretations, Palawan is reconstructed to the northwest of the Late Oligocene–Early Miocene subduction zone along northwest Borneo[16] (Fig. 2 and Supplementary Fig. 1a–g). In contrast, Palawan has also been reconstructed to the southeast of the ancient subduction zone along northwest Borneo[21]. The sensitivity of palaeotidal modelling to Palawan's position was tested by removing an emergent Palawan in base-case palaeogeographic interpretations but the subduction zone position remained consistent (Supplementary Fig. 1h–j).

Tectonics caused significant subsidence in shelf basins in the SCS during the Oligo–Miocene; extension in western and northern basins (Fig. 3a—basins 1–6) contributed to c. >4–5 km total (decompacted) subsidence[18], whereas lithospheric flexure caused up to 8–12 km of subsidence in foreland basins along northwest Borneo[23] (Fig. 3a,b—basins 7–8). Several basins in the western and southern SCS were subject to inversion during the Middle Miocene–Present[16,18]. Variations in the magnitude of tectonic subsidence and inversion, thermal subsidence,

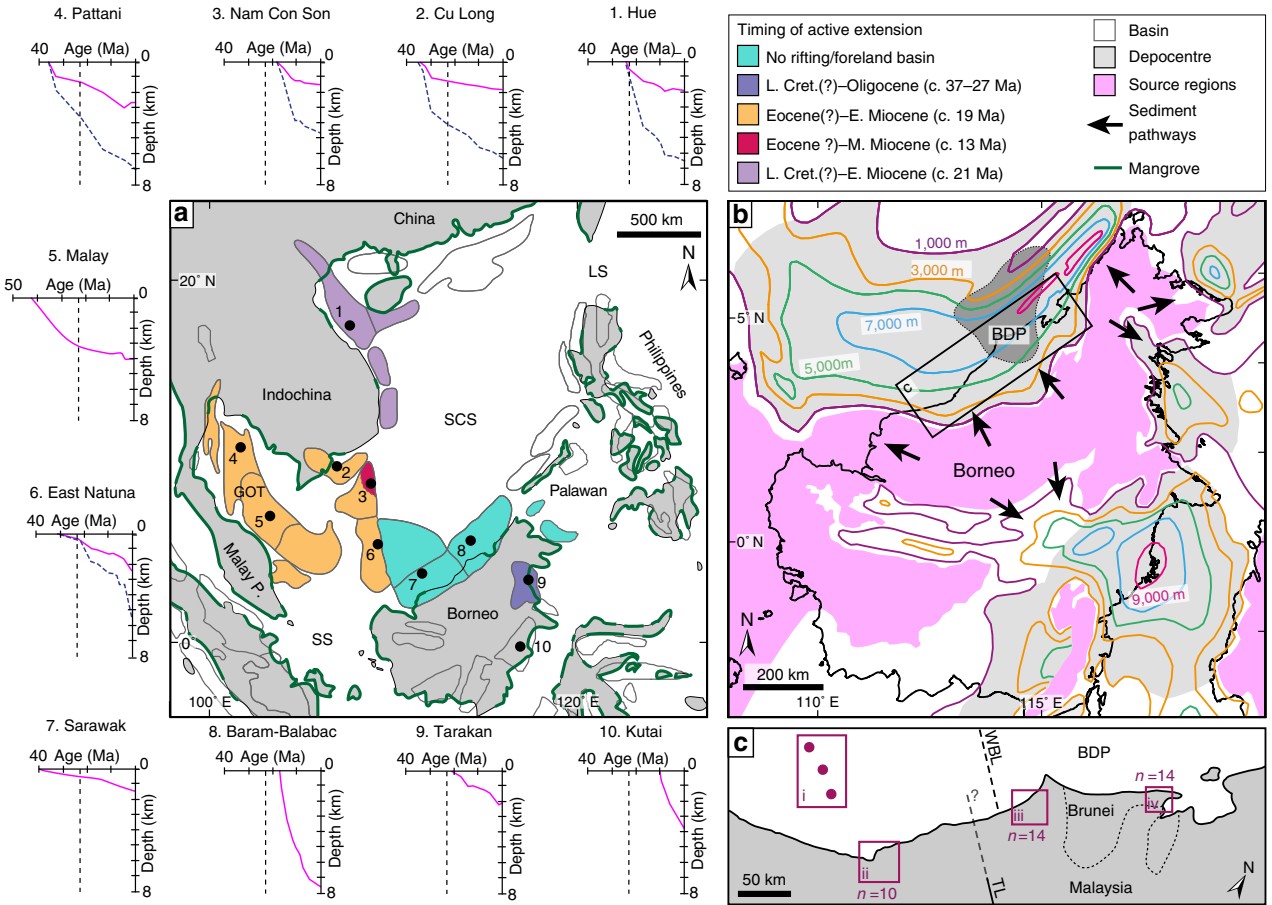

**Figure 3 | Distribution of modern mangroves and Neogene sedimentary basins and their subsidence histories in the SCS region.** (**a**) Timing of active extension in sedimentary basins and representative subsidence histories for 10 basins (1–10; see Methods section)[18,23,58,62–64]. Modern mangrove distribution along coastlines is shown in dark green[1]. Subsidence curves shown are tectonic (purple) and total decompacted (blue dashed). Vertical dashed line indicates the Oligo–Miocene boundary (c. 23 Ma). Cret, Cretaceous; GOT, Gulf of Thailand. (**b**) Neogene sediment thickness map for peripheral Borneo basins[37]. Bold black line is the Borneo coastline. Contours show Neogene sediment thickness every 2,000 m from 1,000 to 9,000 m. (**c**) Northwest Borneo core and outcrop data locations. TL, Tinjar Line; WBL, West Baram Line.

sediment supply and global eustatic sea level caused spatio-temporal changes in coastal-shelf physiography, sedimentation and stratigraphic architecture during the Oligocene–Present. Thermal subsidence after cessation of seafloor spreading (c. Early–Middle Miocene) increased the area of continental shelf (<200 m depth) in the southwest SCS (Gulf of Thailand) through the Middle–Late Miocene (Fig. 2).

**Tidal modelling**. Tidal modelling (see Methods section) of three representative Oligo–Miocene palaeogeographies and palaeo sea levels (at 21, 15 and 6 Ma; see Fig. 2b,d,g) highlight important changes in modelled tidal dynamics during the Oligo–Miocene (Fig. 4; see also Supplementary Figs 2–8). Prevailing tidal range in the central part of the SCS decreased from macrotidal (>4 m) in the Late Oligocene–Early Miocene (Fig. 4a) to mesotidal (>2–4 m) in the Middle–Late Miocene (Figs 4b,c and 5a). The SCS experienced the highest tides (>10 m tidal range) on Earth during the Late Oligocene–Early Miocene. Tidal range in the Gulf of Thailand (western SCS) generally increased from microtidal to low mesotidal during the Oligo–Miocene as the submerged region became wider and deeper (Fig. 4a–c).

Tidal currents along coastlines in the central SCS were generally capable of transporting coarse sand to gravel during the Late Oligocene and Early Miocene (Fig. 4d,e), sand in the Middle Miocene and fine sand to silt in the Late Miocene (Fig. 4f). The maximum bed shear stress of tidal currents along coastlines in the Gulf of Thailand was capable of reworking sand throughout the Miocene (Fig. 4d–f). The percentage of length of coastline subject to macrotidal conditions (Fig. 5a) and tidal currents capable of reworking sand and gravel (Fig. 5b) has decreased since the Late Oligocene to present day.

The Izu–Bonin–Mariana (IBM) arc (Fig. 2) had a significant blocking effect on tides throughout the Miocene, as indicated by a substantial increase in the amplitude and strength of tides when the IBM arc is modelled as emergent (Fig. 5—6 Ma and Supplementary Fig. 8).

**Tidal sediment preservation in the Oligo–Miocene SCS region**. Tidal range and bed shear stress model output has been compared with sedimentological and biostratigraphic data from

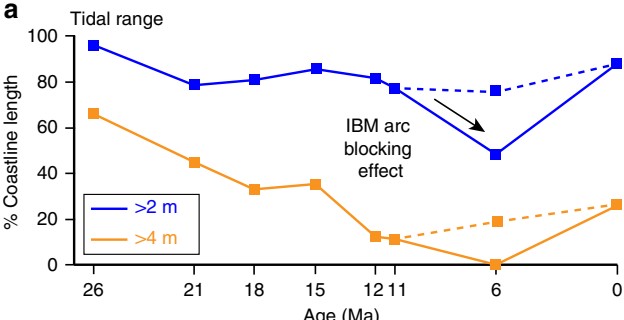

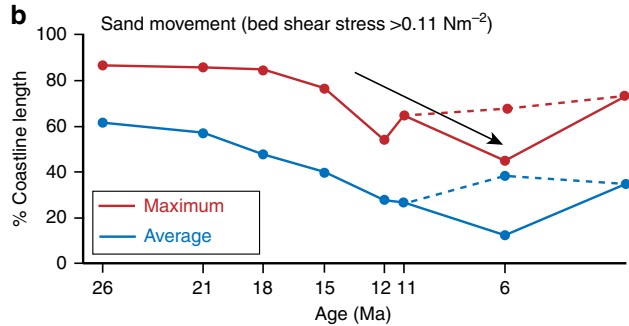

**Figure 5 | Change in modelled tides around the coastline of the SCS since 26 Ma.** Proportion of shoreline length per timeslice (base-case palaeogeography) affected by >2 m (meso-macrotidal) and >4 m (macrotidal) tides (**a**) and tides capable of sand movement (average and maximum modelled bed shear stress) (**b**) (also see Supplementary Figs 3, 4 and 8). Dashed lines show alternative trend for 6 Ma model results with submerged Izu-Bonin-Mariana (IBM) arc.

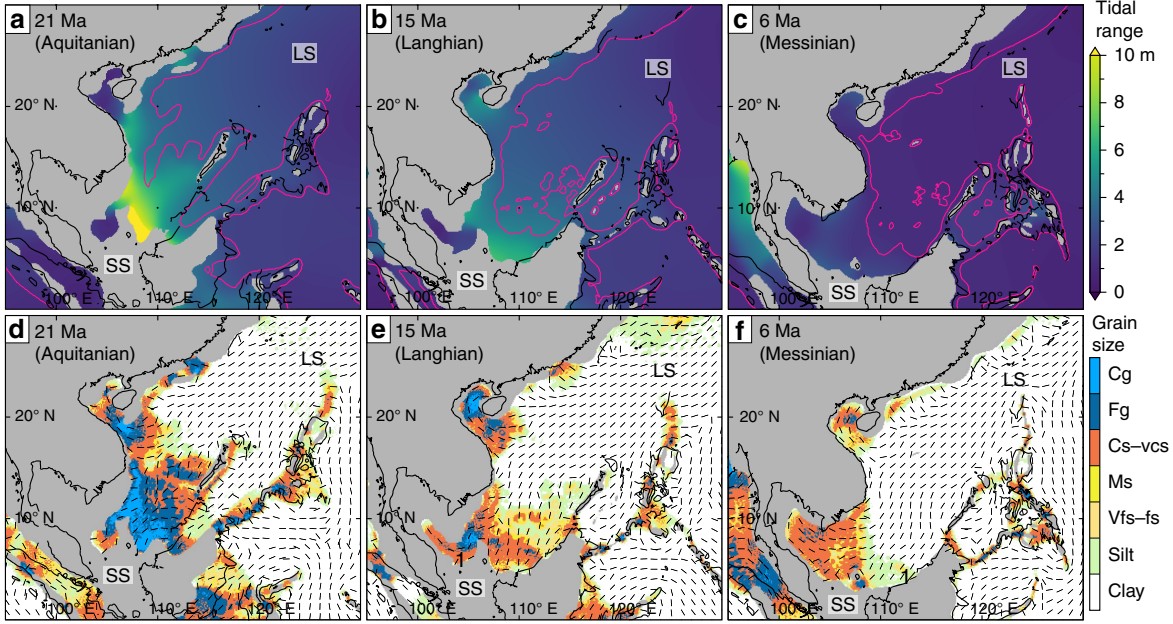

**Figure 4 | Modelled tides in the Oligo–Miocene SCS.** Model results for tidal range (**a–c**) and maximum tidal bed shear stress, plotted as the maximum sediment calibre entrained (**d–f**), for three palaeobathymetric reconstructions: 21 Ma (**a,d**), 15 Ma (**b,e**), and 6 Ma (**c,f**) (cf. Fig. 2b,d,g). The thicker black line (**a–f**) is the reconstructed present-day coastline. The thinner black lines (**d–f**) indicate the direction of maximum tidal bed shear stress. The 200 m palaeobathymetric contour (pink) (**a–c**) is the approximate palaeo-shelf edge. See Supplementary Figs 3–8 for model results of all seven ancient timeslices and sensitivity analyses.

the Oligo–Miocene SCS region. This indicates broad agreement between model results and previous palaeoenvironmental interpretations[18], including the widespread occurrence of mangrove-related facies[24,25]. However, comparing and validating tidal bed shear stress output with sedimentological data[26] (Fig. 6; see Methods section) depends on the grain size availability at the time of deposition, which controls the type and preservation potential of tidal signals in the sedimentary record. In shoreline–shelf depositional systems, grain size availability is controlled by the grain size distribution of sediment supplied to the shoreline through river mouths. Tidal signals will be absent if the available sediment was coarser than the maximum sediment calibre capable of being reworked by the tides. Conversely, if the available sediment was finer than the maximum sediment calibre capable of tidal reworking, then an increase in the size and frequency of tidal bedforms could be expected. A comparison between the model results and the actual stratigraphic preservation for peripheral SCS basins (numbered 1–8 in Fig. 6) is discussed below (see Methods section).

In the Nam Con Son Basin, the Late Oligocene–Early Miocene Dua Formation comprises paralic (coastal–deltaic to shallow marine) coals and mudrocks, commonly containing mangrove palynomorphs[27], and fine-to-medium-grained sandstones (Fig. 6—basin 3), which are interpreted to have been deposited under tidal influence[28]. Modelled tides capable of reworking gravel (Fig. 4e) are consistent with tide-influenced deposition and preservation of decimetre- to metre-scale bedforms in fine-to-medium-grained sandstones (Fig. 6—basin 3).

In the Pattani Basin, Early–Late Miocene mudstone/coal source rocks with common mangrove pollen are interbedded with medium-grained sandstones (Fig. 6—basin 4); the interpreted tide-influenced environments (fluvio–deltaic and mangrove-

vegetated lower coastal plain/lagoons)[29] is consistent with modelled tides capable of entraining coarse sand (Fig. 4e–g) and formation of decimetre- to metre-scale bedforms (Fig. 6—basin 4). In Late Oligocene–Middle Miocene strata in the Malay Basin, paralic mudstones and coals contain abundant mangrove (and freshwater flora) pollen and represent the dominant hydrocarbon source rocks (Fig. 6—basin 5)[17]. Modelled tides were capable of reworking coarse sand (Fig. 4e–g) and preservation of decimetre- to metre-scale bedforms within interbedded medium-grained sandstones (Fig. 6—basin 5). Furthermore, mangrove pollen acmes also occur in Late Oligocene–Middle Miocene strata of the West Natuna and Cuu Long basins[24,27] (Fig. 6—basins 2 and 6), consistent with macrotidal tides capable of reworking sand (Fig. 4a,d). Therefore, the palaeo-Gulf of Thailand, a tectonically controlled regional-scale (100 s of km) embayment, facilitated mangrove colonization throughout the Mio–Pliocene. This was enhanced by a combination of sheltering from direct wave approach and tidal amplification, which compares closely to the modern Gulf of Thailand and Brunei Bay, northwest Borneo.

In northwest Borneo, the Early Miocene Nyalau Formation of the Balingian Province, Sarawak Basin (Fig. 6—basin 7) and the Middle–Late Miocene Lambir and Belait formations in the Baram Delta Province (BDP), a sub-region of the Baram–Balabac Basin (Fig. 3—basin 8), preserve thick and extensive carbon-rich mangrove deposits in tide-influenced coastal–deltaic successions[30]. Nine facies associations (Supplementary Table 1) are arranged vertically into four facies successions (Fig. 6a,b) interpreted to reflect deposition in the following environments: (1) open-coast, storm-dominated delta/shoreface-shelf (FS1); (2) estuary/embayment (FS2); (3) fluvio-tidal channels (FS3); and (4) tide-dominated deltas (FS4). FS2–4 includes very-fine-to-

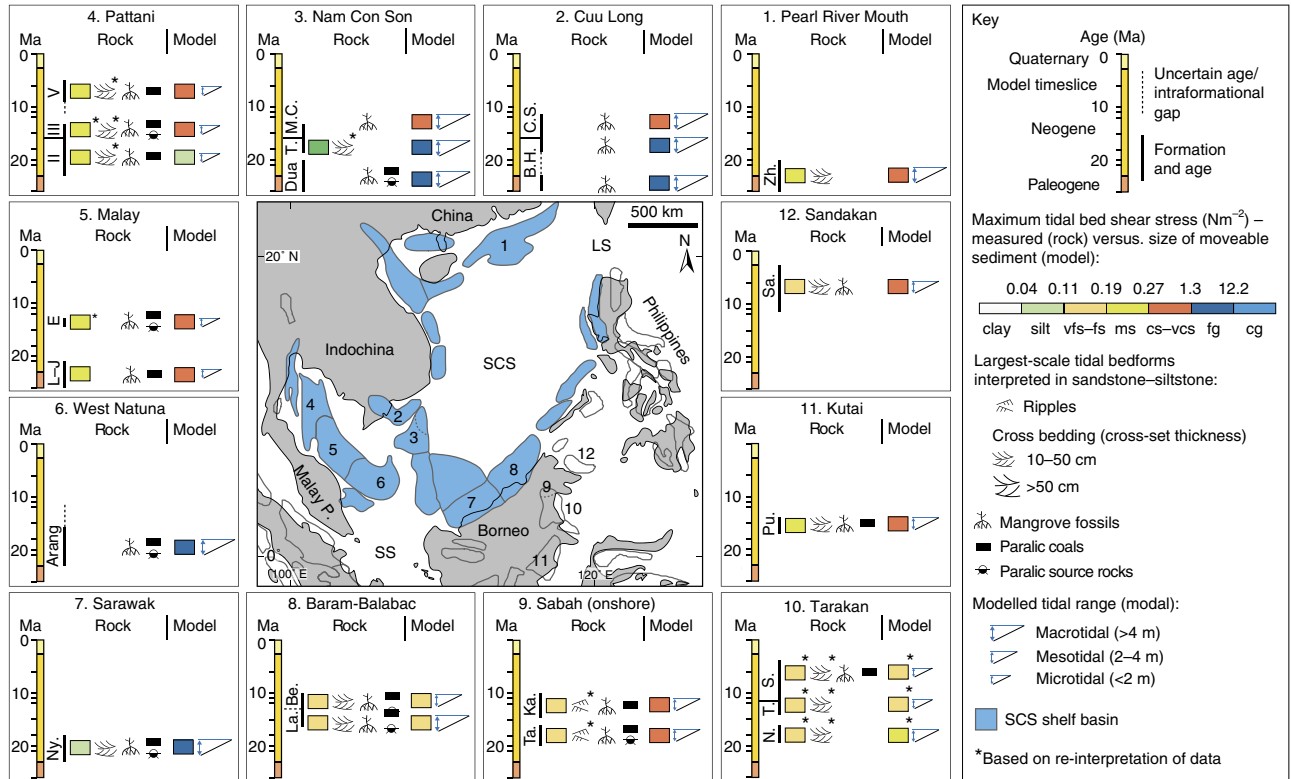

**Figure 6 | Simplified comparison of sedimentological rock data displaying evidence for tidal processes and modelled tides in peripheral SCS and Borneo basins.** Sedimentological parameters are: modal grain size and largest-scale tide-interpreted sedimentary structure in siltstone–sandstone; mangrove pollen acme (typically in mudstones and/or coals); and the presence of paralic (coastal–deltaic to shallow marine) coals and/or source rocks. See Methods section for data sources.

fine-grained sandstone displaying tidal signals, such as bidirectional cross-bedding, abundant reactivation surfaces and impoverished bioturbation[31,32]. Mangrove deposition is confirmed by preservation of abundant mangrove carbargillite microlithotypes[33] and palynomorphs[34], mangrove coals (Fig. 7c), coalified roots and tree stumps (Fig. 7d) and fossilized mangrove leaves. Furthermore, decimetre-scale, branching, mud- and sand-filled, and organic-debris-lined *Thalassinoides*-like trace fossils, may preserve burrow networks typical of the lobster *Thalassina* in modern mangroves and seagrasses and intensely bioturbated mudstones, typical of intertidal mud flats seaward of mangrove forests[35].

The gross depositional environment was a variably tide- and wave-influenced deltaic shoreline, which included mangrove colonization of the intertidal lower coastal-deltaic plain (Fig. 7e)[30,34,36]. Deposition of FS2–4 occurred primarily along an open coastline for the Early Miocene Sarawak Basin (together with FS1) and along an embayed coastline for the Middle–Late Miocene Baram–Balabac Basin (FS1-dominated open coastline environments). Mangrove-carbon rich mudstones (in FS2) were especially well preserved within abandoned fluvio-tidal distributary channels and estuaries[15] (Fig. 7e) and in shallow, wave-protected embayments. This closely resembles preservation of mangrove sediment in modern Brunei Bay. Modelled tides in the Early Miocene Balingian Province were macrotidal and capable of transporting gravel (Fig. 4a,e), which supports the observed decimetre-scale

tidal cross-stratification in very-fine-to-fine-grained sandstones[30] (Fig. 7a,b). Modelled tides in the Middle Miocene BDP were mesotidal–macrotidal (c. 4–5 m) and capable of transporting fine sand (Fig. 4b,f), consistent with decimetre-scale tidal cross-bedding in very-fine-to-fine-grained sandstones (Supplementary Table 1).

**Impact on OC burial in the Oligo–Miocene SCS.** Total organic carbon (TOC) burial in the Oligo–Miocene SCS, which contains a significant component of mangrove OC[17], is evaluated through a series of assumptions regarding the volume of sediment preserved, the volume of hydrocarbons generated, the dominant source rock type and TOC of preserved sediment. We first estimate the amount of OC burial in the BDP (Fig. 3b) since the mid-Miocene (15 Ma) using two approaches: Method 1—volume of hydrocarbons in place (that is, oil and gas that has been proven to be trapped following secondary migration from source rocks); and Method 2—average TOC for the total preserved sediment volume. Detailed calculations are made in the BDP because there is published information on subsidence history[23] (Fig. 3a—basin 8), sediment thickness[37] (Fig. 3b), TOC values of source rocks[38] and hydrocarbon volumes in place[23,39]. Second, we compare the sediment volume in the BDP to the total sediment volume in peripheral Borneo[37] (Fig. 3b) and SCS[40] basins (Fig. 6; Table 1). Third, we estimate the amount of OC burial in peripheral Borneo and SCS basins by assuming that the amount of OC burial per unit

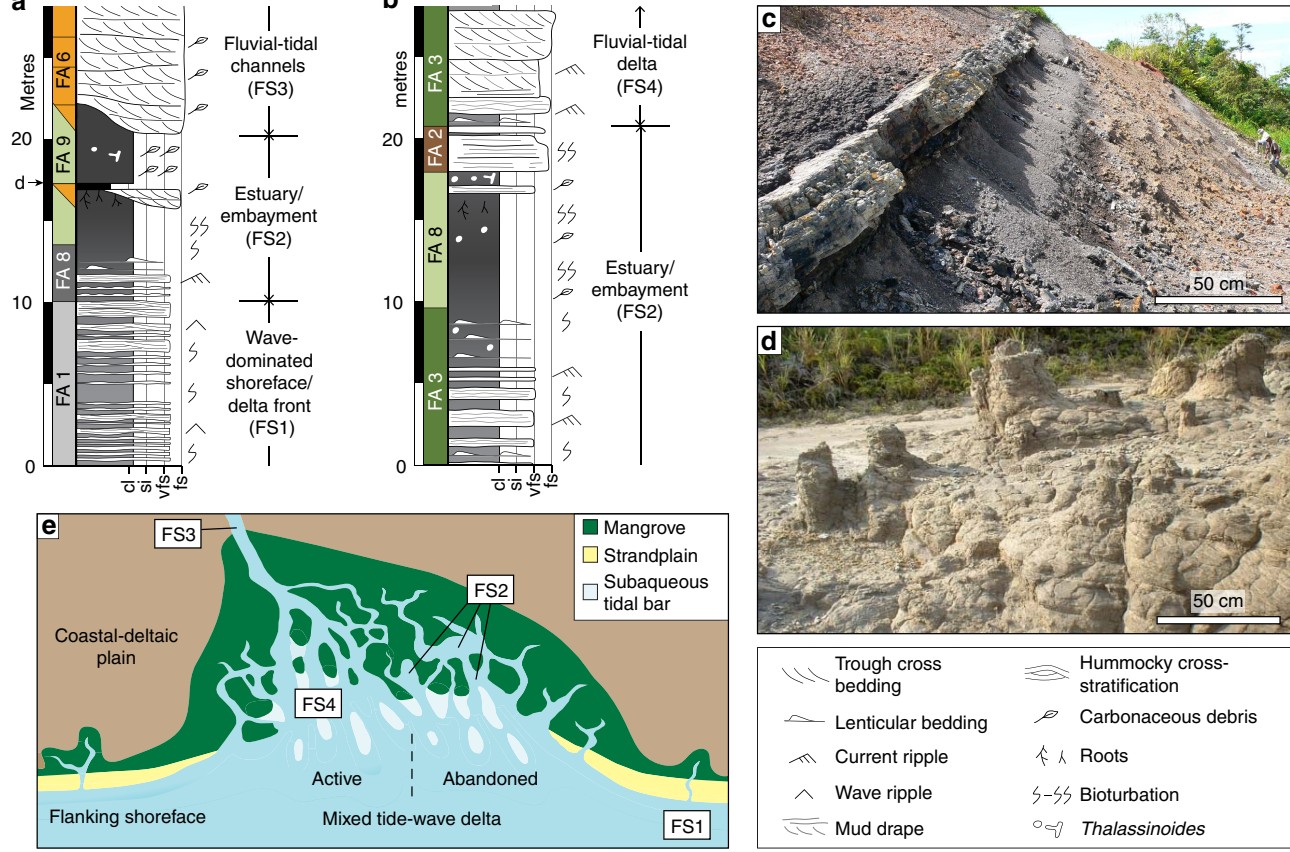

**Figure 7 | Mangrove facies preservation in the Oligocene–Miocene of northwest Borneo.** Examples of mangrove-bearing facies successions (FS1–4): (**a**) in the Early Miocene Nyalau Formation, Balingian Province (Sarawak Basin); (**b**) Middle–Late Miocene Belait Formation, Baram Delta Province (Baram–Balabac Basin). (**c**) Mangrove coal and associated embayment mudstone. (**d**) Preserved mangrove tree stumps, stratigraphically above a mangrove coal (**a**) in the Nyalau Formation. (**e**) Mixed tide-wave depositional model for open deltaic coastlines in the Early Miocene Balingian Province[30] and within coastal embayments in the Middle–Late Miocene Balingian and Baram Delta provinces.

sediment volume in these basins was comparable to that within the BDP since 15 Ma (calculated using Methods 1 and 2) (Table 1). This is assumed because deposition in all these basins occurred in similar tropical, wave- and tide-influenced, coastal-deltaic to deep marine systems, with abundant evidence for mangrove vegetation (Fig. 6), as manifested in the modern Baram, Mekong, Pearl River Mouth, Rajang and Red River deltas. For Method 2, we assume TOC values of 0.05 and 1%, which are comparable to the TOC values measured in intertidal seagrass sediment (c. 0.15–1%) but significantly lower than maximum TOC measured in salt marsh (c. 2–13%) and mangrove sediment (2–37%)[2,5]. We assume relatively low average TOC values to account for: (1) dilution of mangrove OC by reworking and deposition in adjacent coastal environments; (2) variable OC enrichment in different coastal environments; and (3) OC decomposition on geological timescales[41].

In the BDP, exceptional rates of sediment supply have resulted in an average of c. 7 km (compacted thickness; Fig. 3a—basin 8)[23,37] of coastal-deltaic to deep-marine deposition across c. 20,000 km$^2$ since the Middle Miocene (c. 15 Ma) (Fig. 3b). This corresponds to an average sedimentation rate of c. 0.5 mm year$^{-1}$, although at times during the last 15 Ma sedimentation was even higher, keeping pace with tectonic subsidence rates of up to 3 mm year$^{-1}$ (ref. 23). Mangrove OC in coastal–deltaic, shallow-marine and deep-marine sedimentary rocks[23,38] is the dominant source material for substantial volumes of hydrocarbons in the BDP (Brunei[23] and Sarawak[39]); an estimated 7 billion barrels of oil and 19 trillion cubic feet of gas initially in place (before production) (Supplementary Table 2) suggests a minimum OC burial (as trapped oil and gas; Method 1) of c. 1.9 Gt in the BDP since 15 Ma, equivalent to 0.1 p.p.m. atmospheric equivalent $CO_2$ per Myr (Table 1). This represents a minimum estimate because not all of the accumulated carbon would have generated hydrocarbons, not all hydrocarbons have been found, some hydrocarbons have migrated to the surface and shallow occurrences have undergone biodegradation due to freshwater flushing[23].

Using Method 2, with an average TOC of 0.05%, OC burial in the BDP since c. 15 Ma increases to 90 Gt; this is equivalent to sequestration of c. 2.8 p.p.m. atmospheric equivalent $CO_2$ per Myr (Table 1). An average TOC of 0.05% is significantly lower than the average 2.4% TOC ($n = 20$) measured in the western BDP[38]; however, the western BDP sample set was biased towards carbonaceous facies. The paucity of published TOC data in the BDP prevents calculation of a more accurate, average TOC value. However, the high average sedimentation rate in the BDP since c. 15 Ma (c. 0.5 mm year$^{-1}$)[23] suggests a relatively high efficiency for OC preservation due to a decrease in the exposure time to oxygen during burial[41,42]. Since the depth of oxygen penetration in marine sediments is typically 1–10 mm[4], sediments deposited in the BDP were probably exposed to $O_2$ for only a few decades. A maximum average TOC of 1% for sediments deposited in the BDP since 15 Ma may therefore be plausible, especially with an expanded and more productive coastal mangrove biome. An average TOC of 1% would correspond to c. 1,800 Gt OC burial, equivalent to sequestration of 48 p.p.m. atmospheric equivalent $CO_2$ per Myr (Table 1).

The estimated sediment volume deposited in present-day shelf basins of the SCS (Fig. 6; Table 1) since the Oligocene (34 Ma) is $58 \times 10^5$ km$^3$ (ref. 40); only c. 2% ($12 \times 10^4$ km$^3$) of this volume is accounted for by mid-Miocene (c. 15 Ma) to present sedimentation in the BDP. Assuming that the amount of OC buried per unit sediment volume in present-day shelf basins of the SCS was equivalent to that calculated in the BDP (using Methods 1 and 2 and assuming a TOC value of 0.05%), we estimate that between c. 40 and c. 2,000 Gt of OC was buried in SCS shelf basins since 34 Ma (Table 1). This sustained sequestration of c. 3–130 GtC per Myr is equivalent to sequestration of c. 1–60 p.p.m. atmospheric equivalent $CO_2$ per Myr; therefore, burial of mangrove OC in SCS shelf basins could have significantly contributed to an overall decrease in atmospheric $CO_2$ concentration from c. 800 to c. 300 p.p.m. during the Late Oligocene to the Present (34–0 Ma)[43]. The decreased OC burial rate compared to modern mangroves (c. 1,000 GtC per Myr[3]; see Methods section) is due to OC deposition in a range of coastal–deltaic to deep-marine environments, reworking and decomposition on geological timescales. Assuming a maximum average TOC value of 1% suggests burial of c. 41,000 Gt of OC in SCS shelf basins since the mid-Miocene (c. 15 Ma; Table 1), which would account for c. 21% of the total estimated net growth of the global sedimentary OC reservoir during this period (c. 192,000

### Table 1 | Summary of OC burial estimates in the SCS.

| Region | Time period (Ma) | Sediment volume (km$^3$) | Estimate method | Total OC burial (GtC) | Volume of $CO_2$ (p.p.m.) | OC burial rate (GtC Myr$^{-1}$) | $CO_2$ sequestration rate (p.p.m. Myr$^{-1}$) |
|---|---|---|---|---|---|---|---|
| BDP | 0–15 | $12 \times 10^{4,*}$ | 1. | 1.9 | 0.9 | 0.1 | 0.1 |
| | | | 2. TOC = 0.05% | 90 | 42 | 6.0 | 2.8 |
| | | | 2. TOC = 1% | 1,530 | 720 | 100 | 48 |
| SCS shelf basins | 0–15 | $27 \times 10^{5,\dagger}$ | 1. | 42 | 20 | 2.7 | 1.3 |
| | | | 2. TOC = 0.05% | 2,040 | 960 | 128 | 60 |
| | | | 2. TOC = 1% | 40,800 | 19,200 | 2,550 | 1,200 |
| Borneo basins | 0–23 | $34 \times 10^{5,\ddagger}$ | 1. | 54 | 26 | 2.7 | 1.2 |
| | | | 2. TOC = 0.05% | 2,580 | 120 | 126 | 59 |
| | | | 2. TOC = 1% | 51,600 | 24,200 | 2,500 | 1,180 |
| SCS shelf basins | 0–34 | $58 \times 10^{5,\dagger}$ | 1. | 90 | 43 | 2.7 | 1.3 |
| | | | 2. TOC = 0.05% | 4,330 | 2,030 | 128 | 60 |
| | | | 2. TOC = 1% | 86,700 | 40,700 | 2,560 | 1,200 |
| SCS | 0–34 | $70 \times 10^{5,\S}$ | 1. | 109 | 52 | 3.3 | 1.5 |
| | | | 2. TOC = 0.05% | 5,260 | 2,470 | 155 | 73 |
| | | | 2. TOC = 1% | 10,500 | 49,400 | 3,100 | 1,460 |

BDP, Baram Delta Province; OC, organic carbon; SCS, South China Sea; TOC, total organic carbon.
*Sediment volume estimate based on estimates of BDP area and average sediment thickness in refs 23,37.
†Based on ref. 40 and assuming average density (2,060 kg m$^{-3}$) and 82% of SCS sediment mass deposited on shelf since Oligocene.
‡Sediment volume estimate based on ref. 37.
§Sediment volume estimate based on ref. 40.

GtC)[44]. These estimations highlight the significant impact of geological sequestration of mangrove OC to the global carbon cycle during the Neogene, although the precise size of the expanded mangrove biome is uncertain.

## Discussion

Oligo–Miocene tides in the SCS were strongly influenced by rapid and substantial palaeogeographic changes (Fig. 2), including the emergence of the Sunda Shelf, variations in the width and depth of the LS and the position and bathymetry of the IBM arc (Fig. 4 and Supplementary Figs 3,4 and 7). The emergence of the Sunda Shelf was linked to the plate tectonic evolution of Southeast Asia, which created a 'blind' gulf-like basin geometry with uplifted areas on three sides of the SCS and an open ocean connection to the northeast (Fig. 2). Furthermore, during the Oligo–Miocene, the wider and deeper Luzon Strait and ocean connection between the IBM Arc and Japan (Fig. 2) enhanced inflow of tidal energy from the Pacific Ocean compared with the present day (Fig. 4). A decrease in tidal range and tidal bed shear stress through the Miocene reflects a decrease in tidal energy entering the SCS, caused by: (1) the Luzon Strait becoming narrower and shallower due to northward migration of the Philippines and associated volcanic activity[16], and (2) northward movement of the IBM arc.

Thermal subsidence and rising sea levels increased shelf width in the western and southern SCS during the Mio–Pliocene to present day[18]. This would have increased frictional damping of shoaling tides, contributing to decreasing tidal range and strength[26,45]. However, funnelling of tides in palaeobathymetric constrictions produced elevated tidal range and bed shear stresses throughout the Miocene, most notably during the Early Miocene in northwest Borneo (Fig. 6—basin 7) and during the Middle-to-Late Miocene in the Gulf of Thailand (Fig. 6—basins 4–6).

Along tropical coastlines in the Early–Middle Miocene of the SCS, assuming a consistent coastal plain gradient, higher tidal ranges would have increased the intertidal area for mangrove colonization (Fig. 8). Stronger tides would have promoted development of embayments, tidal channels and lagoons, thereby providing greater wave protection for optimum mangrove growth (Fig. 8). Higher tidal range would have also increased tidal flow and salinity in the lower reaches of marine-connected river channels and adjacent tidal channels. This, combined with stronger tides, would have increased the supply of sediment, oxygen and nutrients throughout tide-influenced areas of the lower coastal plain. Together, this would have enhanced mangrove development, productivity and sediment trapping and accretion, leading to increased intertidal OC burial (Fig. 8c)[7,46,47]. Stronger tides would have also increased erosion of tidal channels and maintained their flow capacity. Although modern mangroves are typically found along low wave and tide energy coastlines[12,13], they also occur on open coastlines subject to relatively high tidal energy (for example, Ganges–Brahmaputra[48] and Fly River[49] deltas) and mixed tidal-wave energy (for example, Mekong Delta[13]). If subjected to high coastal energy and degradation, mangroves can rapidly migrate and regenerate[50]. However, higher tidal ranges can promote the stability of tidal channel networks and intertidal vegetation, making mangroves less susceptible to erosion and destruction[46]. This is augmented by the exceptional sediment-trapping capacity of mangrove roots, which dramatically stabilizes sediment in the intertidal zone[7,12].

Preservation of mangrove sediment depends on the interplay between the rates of accommodation space creation and sedimentation[13]. During the Miocene, an overall rise in sea level in the western SCS (Fig. 2) contributed to a long-term ($\geq 10^6$ years)

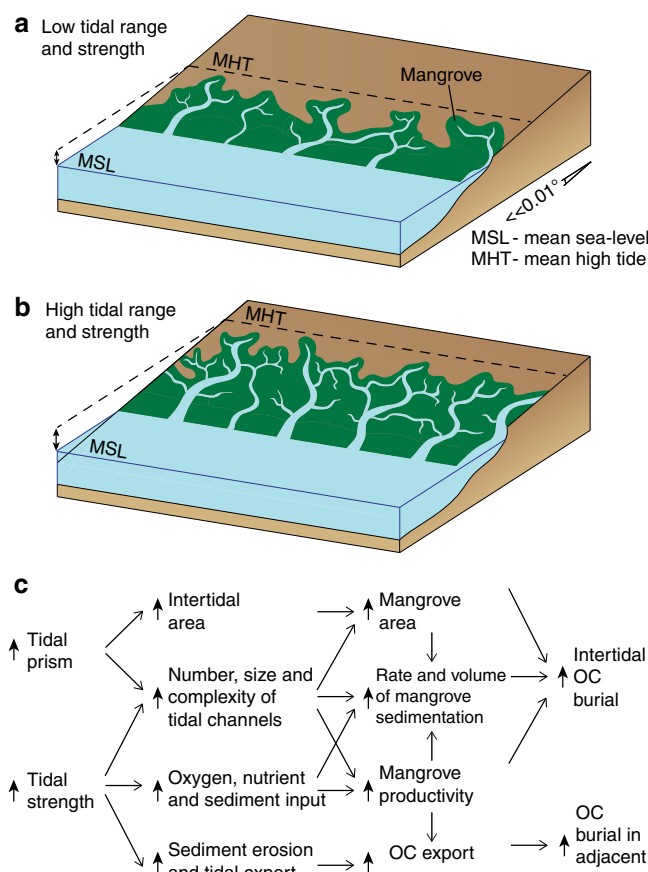

**Figure 8 | The impact of increasing tidal range and bed shear stress on intertidal mangrove colonization.** Schematic changes in tidal channel size, density and complexity and mangrove area distribution between periods of relative low (**a**) and high (**b**) tidal range and strength (bed shear stress), assuming no change in other controlling factors[12]. (**c**) Simplified effects of increased tidal prism and strength on topography and dynamic processes impacting OC burial within the intertidal zone and adjacent environments[7,12].

increase in accommodation space for mangrove accumulation[13] and may have increased OC sequestration independent of changes in tidal dynamics. However, this long-term trend includes many shorter-term fluctuations in relative sea level (RSL), the causes of which include glacio-eustasy[51], sediment supply variations and tectonics. SCS basins display multiple phases of relative shoreline transgression and regression[18] but with distinctive stratigraphic patterns reflecting different tectonic and basin-fill histories (Fig. 3)[18,22]. Shoreline process models[32,52] suggest that tide-influenced coastlines are more common during transgressive phases than regressive phases because shelves are wider (possibly enhancing tidal shoaling and resonance) and estuaries are more abundant (increasing local tidal amplification). However, extensive mangroves and OC burial have occurred along both transgressive[36] and regressive deltaic shorelines[53].

Vertical stacking of tide- and wave-dominated stratigraphic units (facies successions) in the Balingian and Baram Delta provinces (Fig. 7a,b) may have been caused by internal (autogenic) or external (allogenic) forcing. Tide-dominated stratigraphic units may have been preferentially deposited in coastal embayments formed by tectonics or during transgressive phases of allogenic-driven RSL cycles[54]. Alternatively, transitions between relatively tide- and wave-dominated stratigraphy could reflect changes in process dominance related to autogenic-driven

changes in river-mouth position along a tide- and wave-influenced deltaic shoreline. Axial to river mouths, fluvial and tidal processes dominate; tidal currents are funnelled and amplified as they propagate into distributary channels and ebb tides are augmented by fluvial outflow and form elongate tidal bars (Fig. 7e). Lateral to river mouths, fluvial power and tidal amplification is diminished and waves are relatively more important, but tides may still dominate if they are stronger than wave and fluvial processes (Fig. 7e). Lateral changes in process dominance and mangrove distribution are observed in the present-day Ganges–Brahmaputra Delta[32,48]. The eastern Ganges–Brahmaputra Delta is fluvial dominated[48]; however, diminished fluvial sediment supply to the western Ganges-Brahmaputra Delta has meant that tides, supplemented by wave and storm processes, exhibit the dominant control on sediment redistribution across the lower delta plain and delta front[48]. This has formed a dense tidal channel network within which mangroves have proliferated and stabilized intertidal substrates[48]. High sediment supply to the adjacent fluvial-dominated area and effective sediment reworking by tides and storms enhances mangrove sedimentation and optimizes OC burial in tide-dominated areas of the delta.

During RSL changes, mangroves must re-adjust to the changing intertidal area to maintain their capacity to store and bury OC[13,47]. However, previously buried OC may undergo erosion, reworking and oxidation during these periods[5]. This could offset any reduction in atmospheric carbon resulting from preceding OC burial unless reworked mangrove carbon is re-buried before oxidation. RSL rise will increase the accommodation space for mangroves[13] and generally facilitate increased OC burial[47,55], especially when the rates of RSL rise and vegetation growth are balanced[56]. During RSL fall, erosion by fluvial channels may be substantial and can form large (10s of metres deep and 100 s of metres wide) incised valleys[54]. Mangrove-related sediments within incised valleys will mostly be eroded and transported downstream but a component may be re-deposited and stored within the incised valley[57], especially during the subsequent period of RSL rise and valley-fill[54]. Mangrove sediment adjacent to the incised valley may still be preserved during transgressive phases, though oxidation will destroy a proportion of OC in long-lived soil layers.

During both RSL rise and fall, the depth to which preceding mangrove deposits are eroded may be insignificant compared to the thickness of preceding sedimentation. Many peripheral SCS basins have undergone > 5 km subsidence since the Oligocene (Fig. 3a,b): (1) c. 9–12 km in the Baram–Balabac Basin[23]; (2) c. 5–8 km in the Nam Con Son, Cuu Long and East Natuna basins[18]; and (3) c. 12 km in the Malay Basin[58]. High sediment supply during the Oligo–Miocene reflects a combination of factors: (1) deep weathering in a humid-tropical climate; (2) high elevation catchment areas (c. > 2 km), notably the Himalaya, Indosinian, Rajang-Crocker Range (northwest Borneo) and Schwaner (southwest Borneo) orogenic belts[37]; and (3) tectonic uplift in catchment areas (for example, Himalaya, northwest Borneo)[16]. High fluvial sediment supply also contributes to mangrove nourishment by maintaining substrate depth during RSL rise and high OC burial[13,55]. The combination of high subsidence, tidal range, sediment supply and strong tides would have optimized: (1) the development of a potentially vast mangrove biome along tide-influenced coastlines; (2) accretion and burial of OC-rich sediment, both within mangroves and adjacent environments; and (3) lithospheric storage of OC.

Mangroves are one of the most efficient links connecting the atmospheric, biospheric and lithospheric reservoirs of carbon[3,8,9]. During the Oligo–Miocene in the SCS, tidal dynamics optimized the development of carbon-rich mangroves, and high fluvial sediment supply and tectonic subsidence enhanced preservation of mangrove OC. The scale of this mangrove OC burial was a significant component of the global carbon cycle on geological timescales.

## Methods

**Palaeogeographic reconstruction.** Highstand palaeogeographic reconstructions for Southeast Asia at seven timeslices during the Late Oligocene–Late Miocene (Fig. 2 and Supplementary Fig. 1) were generated using the Getech plate model. Reconstructions synthesize diverse published and unpublished sedimentological, stratigraphic and palaeogeographic data[16,18,21,24]. Gross depositional environments are depth-delineated (Supplementary Fig. 1) and the boundaries between these environments are extracted as palaeobathymetric contours and interpolated to form a grid (0.1° resolution) using ArcGIS[59]. Shelf-to-slope (<2,000 m) bathymetry is interpreted from available seismic, sedimentological, lithological, biostratigraphic and palaeogeographic information. Ocean crust bathymetry is calculated by applying an age-depth relationship[60] to a rotated ocean age data set[61], with corrections for sediment cover, sea-level changes and the intrusion history of oceanic seamounts[59]. The detailed reconstructions for Southeast Asia were included in global palaeogeographic reconstructions for each ancient timeslice generated by the Getech Globe project.

Subsidence curves for 10 peripheral SCS and Borneo basins (labelled 1–10 in Fig. 3) are based on ref. 18 (basins 1–4 and 6), ref. 58 (basin 5), ref. 62 (basin 7), ref. 23 (basin 8), ref. 63 (basin 9) and ref. 64 (basin 10).

The maximum variation in global eustatic sea level during the Late Oligocene–Miocene is c. 50 m[51]. Therefore, to test the sensitivity of tidal modelling to palaeobathymetry, we also built tidal models for palaeobathymetric reconstructions for the seven ancient timeslices with sea level 50 m lower than our base-case, highstand palaeobathymetric interpretation (Supplementary Figs 4 and 5). Furthermore, additional tidal modelling was performed to test the sensitivity of model results to two major areas of palaeobathymetric uncertainty. First, to test the sensitivity to the positioning of Palawan relative to the proto-SCS subduction zone along northwest Borneo[22], base-case interpretations place Palawan to the northwest of the subduction zone (Supplementary Fig. 1a–g), whereas we also ran tidal models for palaeogeographic interpretations where Palawan is removed as an emergent feature (Supplementary Fig. 1h–j). Second, the IBM arc is modelled as emergent throughout the Late Oligocene–Miocene (Fig. 1a–g), despite being predominantly submerged in the present day (Fig. 1h). To investigate the blocking effect of an emergent IBM arc on tides entering the Philippine Sea and SCS from the Pacific Ocean (Fig. 1), we ran a tidal model for a Messinian (6 Ma) palaeobathymetric interpretation with the IBM arc submerged to a shallow depth of 10 m (Supplementary Fig. 8).

**Facies analysis.** A detailed outcrop-based sedimentary facies analysis was completed on > 2,000 m of Miocene coastal-deltaic strata in four localities in Northwest Borneo (Fig. 3c): (1) cores (n = 14) from the offshore Balingian Province (Sarawak Basin); (2) outcrops (n = 10) of Nyalau Formation in the onshore Balingian Province[30]; (3) outcrops (n = 14) of the Lambir Formation in northeast Sarawak (BDP, Baram–Balabac Basin); and (4) outcrops (n = 14) of the Belait Formation in northeast Brunei (BDP). This was supplemented by a regional review of published and unpublished sedimentological and stratigraphic data for additional peripheral SCS and Borneo basins (labelled 1–12 in Fig. 6), which focussed on documenting evidence for tide- and mangrove-influenced sedimentation. For each basin, the relevant formations or rock units identified (labelled in Fig. 6) are: (1) Upper Zhuhai (Zh.) Formation, Pearl River Mouth Basin[65]; (2) Bach Ho (B.H.) and Con Son (C.S.) formations, Cuu Long Basin[27]; (3) Dua, Thong (T.) and Mang Cau (M.C.) formations, Nam Con Son Basin[27,28]; (4) Sequence II–IV, Pattani Basin[28,29]; (5) Groups L–J and E, Malay Basin[24,27,29]; (6) Arang Formation, West Natuna Basin[27,66]; (7) Nyalau (Ny.) Formation, Balingian Province, Sarawak Basin[30]; (8) Lambir (La.) and Belait (Be.) formations, Baram Delta Province, Baram-Balabac Basin; (9) Tanjong (Ta.) and Kapilit (Ka.) formations, onshore Central Sabah Basin[67,68]; (10) Naintupo (N.) Tabul (T.) and Santul (S.) formations (Tarakan Basin), all of which contain interpreted tidal sand bars/ridges[63]; (11) Pulau Balang (Pu.) Formation, Kutai Basin[69]; and (12) Sandakan Formation, Sandakan Basin[70].

**Tidal modelling.** Fluidity is a finite element hydrodynamic model that uses unstructured, tetrahedral meshes to maximize computational accuracy and efficiency[45]. Multi-scale, global computational meshes were generated with a finest mesh resolution of c. 10 km in areas of complex bathymetry (for example, steep topography and coastlines). Fluidity does not permit large-scale flooding/drying, therefore, there must be a minimum depth along the coastline to prevent the free surface from intersecting the bottom surface as it propagates[45]. Simulations represent full astronomical tidal forcing for 3 months of simulation time with a spin-up period of five days. Outputs are the amplitude of constituent tidal components, tidal range and the magnitude and direction of average and maximum tidal bed shear stress. Tidal range is calculated as the difference between the maximum and minimum free surface heights, which approximately equates to the maximum spring tidal range.

Fluidity has been extensively validated for tidal modelling in modern and ancient settings[26,45], including the SCS (Supplementary Fig. 2). The pattern and magnitude of $M_2$ and $K_1$ tides in the SCS compare favourably to models that include data assimilation: (1) global tidal models ATLAS TPXO8 and FES2012, which both used structured meshes with c. 20 km (1/6°) and c. 7 km (1/16°) resolution, respectively[19]; and (2) a regional tidal model OTIS[19] that used a structured mesh with c. 4 km (1/30°) resolution. For the $M_2$ tide, Fluidity underpredicts the amplitude in the Philippine Sea (Supplementary Fig. 2), most likely due to a coarser mesh resolution. For the $K_1$ tide, Fluidity overpredicts the amplitude slightly ( < 0.1 m) in the western SCS (Supplementary Fig. 2), most likely because insufficient energy is damped by frictional drag along the bottom surface, due to a coarser mesh resolution and smoothed bathymetry, and lack of internal drag[19]. For ancient simulations, internal drag is not included because the buoyancy frequency (a function of water column density gradient) is unknown.

**Carbon burial estimates.** Estimates of the total amount and rate of OC burial, and equivalent concentration of $CO_2$, in the BDP since the Middle Miocene (c. 15 Ma), are derived by two methods: Method 1—using estimates of total in-place petroleum resources in hydrocarbon fields in Brunei[23] and Sarawak, Malaysia[39] (Supplementary Table 2); and Method 2—by estimating the sediment volume deposited since c. 15 Ma, using the area of the BDP defined in ref. 23 and sediment thickness map in ref. 37, and assuming average TOC values for the total sediment volume of 0.05 and 1% (Table 1). To estimate the total amount and rate of OC burial, and equivalent concentration of $CO_2$, in peripheral Borneo and SCS basins throughout periods of the Oligocene–Present, we first assumed an equivalent amount of OC burial per unit sediment volume to that calculated in the BDP since the Mid-Miocene (using Methods 1 and 2), and second calculated various estimates of total sediment volume across the region during the Late Oligocene–Present (Table 1). These include the preserved sediment volume in peripheral, present-day shelf basins in the SCS (Fig. 6) since the Middle Miocene (c. 15 Ma) and Oligocene (c. 34 Ma) and the entire SCS since the Oligocene (c. 34 Ma); these are based on Table 1 in ref. 40, assuming a consistent average sediment density of 2,060 kg m$^{-3}$ and that 82% of sediment mass deposited in the SCS since the Oligocene (c. 34 Ma) was deposited in present-day shelf basins[40]. Estimated sediment volume deposited in peripheral Borneo basins (Table 1) during the Neogene is based on ref. 37.

The rate of OC burial in modern mangroves bordering the SCS (c. 1,000 Gt OC per Myr) was calculated using estimates of mangrove area in countries bordering the SCS[1] and assuming the mean global burial rate for soil carbon in mangroves[3].

The proportion of hydrocarbons dominantly sourced by terrestrial- and mangrove-dominated OC (paralic source rocks) in basins in the SCS region was estimated based on the recoverable hydrocarbon volumes and dominant source rocks in ref. 17.

**Data availability.** The palaeogeographic and tidal modelling data sets generated during and/or analysed during the current study are not publicly available due to confidentiality restrictions but are available from the corresponding author on reasonable request and with the permission of Getech. The sedimentary and OC data sets are available from the corresponding author on reasonable request.

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

## Acknowledgements

This work was funded by a NERC PhD scholarship (to D.S.C.). Fieldwork was supported by Shell International Exploration and Production (to D.S.C. and H.D.J.), Brunei Shell Petroleum Co. Ltd. (to A.R.D.) and the University of Malaya grant RP031A-15AFR (to M.H.A.H.). We also acknowledge the support of Getech (to D.S.C., A.A. and P.A.A.) as well as Imperial College's Grantham Institute (to M.D.P.) and High Performance Computing Service. We also thank M. van Cappelle and C. Dean for fieldwork assistance and many other geoscientists for stimulating mangrove discussions during field excursions to northwest Borneo. D. Lunt and B.K. Levell are thanked for valuable comments.

## Author contributions

Field data were collected and analysed by D.S.C., M.H.A.H. and H.D.J. Model experiments were conducted by D.S.C. and A.A., with contributions from J.H., P.A.A. and M.D.P. The manuscript and figures were drafted by D.S.C., P.A.A. and H.D.J., with contributions from the other authors.

## Additional information

**Competing interests:** A.A. is funded by Getech who provided palaeobathymetric data for this work. The remaining authors declare no competing financial interests.

