## [Peer Review File · Nature Communications]

Reviewers' Comments:

Reviewer #1 (Remarks to the Author)

This is an interesting manuscript that links mangrove forest development in the South China Sea with the petroleum deposits in that region and to historic changes in atmospheric CO₂. It could be a highly cited study if published, supporting the proposal of the importance of coastal ecosystems to productivity of the coastal zone (Duarte et al. 2005) and thus regulation of CO₂ in the atmosphere.

However, the manuscript is difficult to understand in places. There are a lot of facts presented but sometimes the interpretation and significance are not clearly presented.

- L12 Tidal range has been used by others as a proxy for the distribution of mangroves (see Woodroffe et al. 1992 or Lovelock et al. 2015), but it is not the only determinant of areal coverage - topography also has to be suitable. This is covered in line 75, but late in the manuscript.
- L13 "while tidal currents influence rate of sediment.... and OC burial" this sentence is a very sweeping generalization. Tides deliver sediments which facilitate OC burial, but sediment supply is also critical.
- L31 - Duarte et al. 2005
- L40 - decreaseover time?
- L46 - is the significance of point (3) that hydrodynamic "energy" declines over time? Mangrove development is usually associated with relatively low energy conditions.
- L49 - is the meaning of point 4 that hydrodynamic "energy" is maintained through the Miocene? Is the significance of this that sediment can be continuously supplied and deposited in mangrove habitats?
- L66 - "active extension in the Gulf of Thailand" What does this mean?
- L76 - Fig 2 b-c seems like a very basic concept to take this space
- L77 - 82 While increased tidal range on gently sloping coastal plain would increase size of the potential mangrove habitat, increases in sedimentation and OC burial would be dependent on the size of accommodation space (see Woodroffe et al. 2016). Increasing relative sea level rise increases accommodation space; otherwise sedimentation declines over time as mangrove habitats reach the highest elevation in the tidal frame.
- L99 - is there a better reference than an unpublished dissertation?
- L112 - could these also be consistent with seagrass beds?
- L115 - could be consistent with low intertidal mud flats. Modern analogues are below mean tide and thus often seaward of mangrove forests.
- L117-119 citations to Woodroffe 1982 or other work that describes development of "big swamp" phase in northern Australia might be an appropriate here.
- L128 - 132 I was not convinced by this set of statements because there is a lack of process based information (or links to coastal processes) to understand the conclusions. There needs to be a clearer articulation of the hypotheses for accumulation of organic rich sediments.
 - o Why are transgressive phases associated with low fluvial supply?
 - o Why would tide processes relative to wave and fluvial processes enhance OC deposition? In modern settings fluvial environments have higher OC burial (see Donato et al.)
 - o Waves lead to resuspension which is important for sedimentation (Lovelock et al. 2014);
 - o Fluvial material is essential for sedimentation;
 - o Where sediment is not abundant organic matter accumulation in mangrove sediments has been associated with relatively slow rates of SLR (see McKee et al. 2007); i.e. changes in accommodation space would enhance sediment deposition.
- L133 onwards - this section needs to be rewritten to make it clearer what is being proposed here
- L139 - is this 1.9 Gt the amount that leads to the 900 ppm draw down mentioned in the Introductory section. Why is the 900 ppm draw down not mentioned here?
- L142 - 90 Gt - is this the 900 ppm draw-down value? Rather than get caught up in the

limitations the authors need first to state the case clearly

- 0.05% TOC is comparable to some values from intertidal mud flats and seagrass beds in addition to fluvial-influenced mangrove - but on the low end compared to most mangrove sediments (2 - 5% OC). Or does 0.05% arise from decomposition over very long time scales?
- L156-158 what is the significance of this statement? Does it mean that decomposition has been significant, or that rates of OC accumulation is likely higher than estimated?
- L158 Citation of 5 and 9 suggests that the authors multiplied an area of mangrove forest from one reference x OC burial number from Alongi 2012. Which value from Alongi was used? Mean, median, min, max?
- L159-161 The significance of these statement and much of the whole paragraph is not clear. This is the section where the authors scale up to the whole region? Is there a tautology here? The authors seem to use a higher TOC content to estimate the size of the OC reservoir but then say the estimates don't include mangrove productivity.
- L169 - 170 - what is the significance of this statement?
- Blair and Aller 2013 is a good reference to consider for OC diagenesis
- References. 19,20,21 are these available?

Reviewer #2 (Remarks to the Author)

Tectonically-driven changes in tidal dynamics drive mangrove carbon sequestration during the Oligo-Miocene

In my opinion this paper has the potential to become suitable for Nature communications, as it brings a very interesting message forward. However, in my opinion, it is not suitable in its present form, for the following two main reasons:

- the paper is not written so that it can be easily read by a broad audience. If you are not into Palaeogeography, you basically need to go to the web to be able to read the text Same for location names; makes you miss a proper map. In my opinion, every paper in Nature communications should be easily readable by people outside the research field. But this is a trivial point, as it can be easily improved by some rewriting
- The main problem that I have with the paper is that it does not get clear to me to which extent having an increased tidal range / tidal dynamics is essential. Figure 1a, 1b and 1c, shows that part of the land gets flooded over time. This suggest that mangroves would gradually shift inward over time, and sequester carbon in the process. So the question I would like to see explicitly answered is: to what extent might this process alone explain the carbon sequestration by mangroves, regardless wether there was a changes in tidal range or not? How much does an altered tidal range add on top of this? The paper states " The elevated tidal range and tidal bed shear stress optimized the development of 18 a vast mangrove biome and associated carbon-rich sediments bordering the SCS. Over 19 geological time scales (106 years) ", suggesting the change in tidal range to be of crucial importance. But as far as I understand, there is not comparison to the situation not accounting for a change in tidal range. Please be aware that I am not a specialist in this topic, and if I here raise a question that is obviously not relevant, please ignore this comment. But as is, it was not clear to me.

Hope this feedback allows the authors to submit a revised manuscript, as the topic is very interesting

Reviewer #3 (Remarks to the Author)

This is an interesting paper that links Oligocene-Miocene paleoenvironmental and tidal reconstructions around the South China Sea to the existence of a large productive mangrove biome and addresses impacts to the global carbon cycle. The tidal modeling component is emphasized in the paper. I would have liked to have seen more emphasis placed on supporting the paleoenvironmental reconstructions and the distribution of mangroves through time. Most of the results presented appear to be based on gross estimates and assumptions. For example, much of

the text focuses on the Baram Delta Province. The authors appear to apply information from that system to four other large deltas in the SCS to scale carbon sequestration numbers to the larger basin. It is not clear why the TOC values for the Bengal Fan should be applied to the SCS. Overall I found that the main point of the study, that the mangrove biome in the SCS contributed significantly to the global carbon cycle on geological timescales, could be true, but is not supported well with the information presented here.

Line 12: It is not only tidal range that determines the areal extent of the mangrove habitat. The substrate depth is just as important. In addition, tidal currents are not the primary determinant on the rate of mangrove sediment deposition because the rate of sea-level rise is extremely important in this regard.

It would be nice if the authors would include the geologic record of atmospheric CO₂ concentrations so the reader can put the stated "it could have contributed to a decrease in atmospheric CO₂ concentrations of 900 ppm since the mid-Miocene" in perspective to other fluctuations in CO₂. Over what period of time, exactly, do the authors think the reduction occurred over? This is important information to include in the first paragraph. Mid-Miocene to present is very broad.

The link between the tidal modeling and the impact on the mangrove biome is not supported well with hard data. It is not clear to this reader exactly how increased tidal energy promotes increased preservation of mangroves (Line 74)

Extended data figure 1. This is an important figure, but difficult to digest, because environmental classifications are vague and colors used are not distinctive. I cannot tell the difference between Lacustrine and Coastal on the maps. I cannot find the intertidal color on the maps. The continental slope is only defined as a water depth, which is incorrect because it is a physiographic feature. There are areas on the map with that color that are not part of the slope (but again difficult to tell with the similarity in colors). What does "coastal" mean? It seems like coastal should be indicated everywhere there is a coast, but that is not the case.

On line 253, "a -50 m sea level transgression" is odd phrasing to me. -50 m implies sea-level fall. Essentially, the palaeobathymetric uncertainty is the entire depth of the shallow shelf? That seems high when the focus of the paper is on Mangroves that live in shallow water. I might be missing something here.

I don't understand what >2000 is referring to on line 256. Is this >2000 meters of mangrove-bearing, Miocene coastal-deltaic strata or >2000 samples, or something else. What is the spatial distribution of the samples? How did you interpolate between observations?

Line 161. Is that the global sedimentary OC reservoir that is being referred to? I'm a bit lost here. Why is the TOC percentage from the Bengal Fan being applied to the SCS on lines 162 and 163?

Reviewer #4 (Remarks to the Author)

Dear authors,

The manuscript "Tectonically-driven changes in tidal dynamics drive mangrove carbon sequestration during the Oligo-Miocene" provide insights into the importance of carbon storage by mangroves over geological timescales, from the Oligocene to present. The topic is exciting and novel, and the approach seems sound although my lack of expertise on modelling precludes a comprehensive judgement of this manuscript. My main concern relies on the number of assumptions required to reach the conclusions raised by the authors, such as the uncertainties of the models, and the estimates of volume of organic deposits and hydrocarbons.

Major comments:

1. The authors discussed the importance of tidal range and shear bed stress in driving OC storage and mangrove area, but 'little' or none mention to how changes in sea level rise may impacted OC storage. Thus, coastal geomorphology (slope), which vary across the SCS, may have a significant effect on the area of mangrove. Indeed, the manuscript only focusses in OC storage, but the e.g drop in sea level (over glacial/inter-glacial periods) would expose the OC-rich soil mangrove to oxygen, increasing CO₂ emissions (ppm) and ultimately compensating/enhancing CO₂ due to changes in mangrove area or tide bed shear stress. Please discuss how all above may influenced OC storage (balance) over geological scales.

2. L22-23. Estimates of atmospheric CO₂ concentration (ppm) over the Miocene point to values around 1000 ppm, but I don't follow how the TOC sequestration by mangrove could led to a decrease in atmospheric CO₂ concentrations of c. 900 ppm. Please explain how you reach this conclusion and if it does make sense in a broader context.

Minor comments:

L14: 170 g OC m⁻² yr⁻¹ instead of 170 gC m⁻² yr⁻¹

L16-18: this hypothesis is not supported in the sections below. Please clarify

L18. Elevated tidal range will increase the areal extent of mangrove area, but elevated tidal bed shear stress may cause sediment erosion rather than accumulation of carbon-rich sediments. seems contradictory to me. Please rephrase or reconsider this hypothesis.

L63: "...due to northward migration..." instead of "...due northward migration of..."

L79-82: Increase sediment exchange/tidal bed shear stress would probably resulted in OC export and remineralization, overall decreasing OC accumulation. Please clarify.

L120: add reference to the meaning of BDP here

L145: replace TOC for OC.

Response to Referees:

- **Reviewer comment (line numbers refer to original submission)**
 - **Author response (line number refer to revised submission)**

Reviewers' comments:

Reviewer #1 (Remarks to the Author):

This is an interesting manuscript that links mangrove forest development in the South China Sea with the petroleum deposits in that region and to historic changes in atmospheric CO₂. It could be a highly cited study if published, supporting the proposal of the importance of coastal ecosystems to productivity of the coastal zone (Duarte et al. 2005) and thus regulation of CO₂ in the atmosphere.

However, the manuscript is difficult to understand in places. There are a lot of facts presented but sometimes the interpretation and significance are not clearly presented.

- L12 Tidal range has been used by others as a proxy for the distribution of mangroves (see Woodroffe et al. 1992 or Lovelock et al. 2015), but it is not the only determinant of areal coverage - topography also has to be suitable. This is covered in line 75, but late in the manuscript.

- We have extensively modified our discussion on the control of tides on mangrove distribution, productivity, sediment accretion and OC burial (L305-337 and Fig. 8). We have also added the references Woodroffe et al. 1992 (ref. 14) and Lovelock et al. 2015 (ref. 54).

- L13 "while tidal currents influence rate of sediment.... and OC burial" this sentence is a very sweeping generalization. Tides deliver sediments which facilitate OC burial, but sediment supply is also critical.

- We agree and explicitly mention the importance of sediment supply in L15-22, L35-37, L324-330 and discuss the impact of sediment supply (and other factors) on mangrove sediment preservation during RSL changes in Lines 373-387. Refer also to Fig. 8.

- L31 - Duarte et al. 2005

- This reference has been added (ref. 2)

- L40 - decreaseover time?

- We have modified the nature and structure of our description of tidal model results, employing a similar phrasing (e.g. ...tidal range decreased from... in the Late Oligocene, to ...) (L122-128)

- L46 - is the significance of point (3) that hydrodynamic "energy" declines over time? Mangrove development is usually associated with relatively low energy conditions.

- Bed shear stress is used as a proxy for hydrodynamic 'energy' and therefore the capacity for reworking of sediment by tides. This would affect: (1) the impact on preserved coastal-shelf sedimentary record (L142-210 and Fig. 6), and (2) the

strength of inflows carrying nutrients and oxygen, affecting mangrove productivity (L309-312 and Fig. 8). We agree that mangroves usually occur along relatively low-energy coastlines, especially with respect to wave energy. However, we also suggest that mangrove development occurs where tides are relatively strong (for reasons discussed in L307-315, Fig. 8c), and that tides can develop and regenerate along high-energy modern coastlines (L318-319, and cited references).

- L49 - is the meaning of point 4 that hydrodynamic "energy" is maintained through the Miocene? Is the significance of this that sediment can be continuously supplied and deposited in mangrove habitats?
 - Yes, this is one of a number possible effects of stronger tides (higher hydrodynamic 'energy') on mangrove development considered in L307-315 and Fig. 8.
- L66 - "active extension in the Gulf of Thailand" What does this mean?
 - We have modified the text to show that shelf expansion coincided with post-rift thermal subsidence. This was diachronous across the western SCS (c. Middle Miocene in the Gulf of Thailand). Rifting itself was not a significant part of shelf expansion, rather it pre-dates (L298-299).
- L76 - Fig 2 b-c seems like a very basic concept to take this space
 - We have constructed a more comprehensive (but still greatly simplified) cause-and-effect flow chart in Fig. 8c, and have made simplified graphical block diagrams to help the readers visually understand the main changes we invoke regarding biogeomorphology in intertidal areas, as a result of increased tidal range (prism) and tidal bed shear stress (hydrodynamic 'energy').
- L77 - 82 While increased tidal range on gently sloping coastal plain would increase size of the potential mangrove habitat, increases in sedimentation and OC burial would be dependent on the size of accommodation space (see Woodroffe et al. 2016). Increasing relative sea level rise increases accommodation space; otherwise sedimentation declines over time as mangrove habitats reach the highest elevation in the tidal frame.
 - We now include a more complete discussion on the controls of RSL and tectonics in creating accommodation space for mangrove sedimentation and OC burial (L324-335 and L360-387), including reference to Woodroffe et al. 2016 (ref. 13)
- L99 - is there a better reference than an unpublished dissertation?
 - The author of the dissertation (Abdul Razak Damit) is a long-standing collaborator in aspects of our Borneo outcrop work and is now included as a co-author; the dissertation is available on request. Co-authors M.H.A.H., H.D.J. and P.A.A. are co-authors on a paper in press containing relevant data on Brunei Bay (doi: 10.1144/SP444.12).
- L112 - could these also be consistent with seagrass beds?
 - L192-194 – the possibility of *Thalassoma* in seagrasses is acknowledged.
- L115 - could be consistent with low intertidal mud flats. Modern analogues are below

mean tide and thus often seaward of mangrove forests.

➤ L195-196 – this possibility is now acknowledged.

• L117-119 citations to Woodroffe 1982 or other work that describes development of "big swamp" phase in northern Australia might be an appropriate here.

➤ Line 197-199 - Woodroffe et al 1985 (ref. 36) is now included as a reference

• L128 - 132 I was not convinced by this set of statements because there is a lack of process based information (or links to coastal processes) to understand the conclusions. There needs to be a clearer articulation of the hypotheses for accumulation of organic rich sediments.

➤ The hypotheses on accumulation of organic rich sediments is now expanded and revised in L197-205 and L324-359), including possible autogenic vs. allogenic controls on the vertical partitioning of tide- and wave-dominated stratigraphy (L338-359).

• Why are transgressive phases associated with low fluvial supply?

➤ This is no longer included as a fundamental component of process changes in transgressive vs. regressive phases (L340-342).

• Why would tide processes relative to wave and fluvial processes enhance OC deposition? In modern settings fluvial environments have higher OC burial (see Donato et al.)

➤ We suggest there is a higher possibility of higher and stronger tides during transgressive phases (L333-337) based on published models of changes in processes during sea level changes (refs 32, 52). Higher and stronger tides would potentially lead to expanded and more productive mangroves (L305-315), and therefore possibly lead to enhanced OC deposition. However, we assume a consistent sediment supply and coastal plain gradient, and acknowledge the importance of accommodation space and sediment supply in creating the space for and allowing preservation of OC (e.g. L324-330 and 373-387).

• Waves lead to resuspension which is important for sedimentation (Lovelock et al. 2014);

➤ We no longer discuss the possibility of higher wave energies, however, we do discuss the link between stronger tides and increased sediment input (through reworking/resuspension; L309-314).

• Fluvial material is essential for sedimentation;

➤ We make reference to this in our discussion on the Ganges-Brahmaputra as an analogue for a mixed tide-wave delta system with extensive mangroves, and how sediment supply optimizes sedimentation and burial of OC in mangroves adjacent to fluvial outflow (L352-359).

• Where sediment is not abundant organic matter accumulation in mangrove sediments has been associated with relatively slow rates of SLR (see McKee et al. 2007); i.e. changes in accommodation space would enhance sediment deposition.

➤ L309-314- McKee et al. 2007 is now referenced (ref. 47).

- L133 onwards - this section needs to be rewritten to make it clearer what is being proposed here
 - L211-235 - Introductory paragraph to 'Impact on OC burial in the Oligo–Miocene SCS' section explicitly states aim, different methods, assumptions and justifications for assumptions.
 - L236-264 and L265-284 – Sub-sections to split calculations based on regions included
 - We have also included Table 1 to show the various estimates per method for different regions and time periods

- L139 - is this 1.9 Gt the amount that leads to the 900 ppm draw down mentioned in the Introductory section. Why is the 900 ppm draw down not mentioned here?
 - Revised on L242-247. 1.9Gt is equivalent to only 0.9ppm CO₂ (Table 1). We then expand on why this is a minimum estimate (L247-250).
 - We no longer refer to a drawdown figure of 900ppm, rather we explicitly state that the ppm volume stated are the equivalent amount of CO₂ to the estimates of buried OC (e.g. L20-22).

- L142 - 90 Gt - is this the 900 ppm draw-down value? Rather than get caught up in the limitations the authors need first to state the case clearly
 - The methodology is now clearly stated in L212-226 and then the limitations are discussed (L226-235).

- 0.05% TOC is comparable to some values from intertidal mud flats and seagrass beds in addition to fluvial-influenced mangrove - but on the low end compared to most mangrove sediments (2 - 5% OC). Or does 0.05% arise from decomposition over very long time scales?
 - 0.05% is simply a very conservative estimate of average TOC for the total sediment volume (which includes mangrove sediment and sediment deposited in environments only partly or not influenced by mangroves). Justification for why the value is consider conservative is given in L229-235.

- L156-158 what is the significance of this statement? Does it mean that decomposition has been significant, or that rates of OC accumulation is likely higher than estimated?
 - Yes, the significance is to illustrate that average OC accumulations rates across a coastal-shelf to deep-marine basin will be less, and decomposition effects will also decrease OC preservation (L232-235 and L256-262). Note that we are estimating total OC burial for the entire range of ancient sedimentary environments, not just mangroves (L226-229).

- L158 Citation of 5 and 9 suggests that the authors multiplied an area of mangrove forest from one reference x OC burial number from Alongi 2012. Which value from Alongi was used? Mean, median, min, max?
 - The mean value from Alongi 2012 was used. The relevant details are now expanded in the methods L491-493.

- L159-161 The significance of these statement and much of the whole paragraph is not clear. This is the section where the authors scale up to the whole region? Is there a

tautology here? The authors seem to use a higher TOC content to estimate the size of the OC reservoir but then say the estimates don't include mangrove productivity.

- We have expanded the explanation of the methods of estimation (including the choice of TOC values) in L229-235 (see also Methods L472-490). Table 1 also includes a comparison of the sediment volume estimates and of all TOC and CO₂ estimates, and relevant references.
 - We have separated estimates based on region to improve clarity.
 - The significance is expressed in terms of the total estimate growth in the sedimentary OC reservoir during the mid-Miocene to present-day (L279-282; ref. 44) and possible importance of OC burial in SCS to global CO₂ decrease during this period (L272-276; ref. 43).
 - The estimates do not account for a change in the size of the mangrove biome and therefore do not directly account for an increase in mangrove OC burial. But we state that we expect the OC to contain a significant proportion of mangrove OC (L59-62 and L212-215), as indicated by source-rock analysis of extensive petroleum deposits in the region (ref. 17; Methods L494-496).
- L169 - 170 - what is the significance of this statement?
 - This statement has been deleted. Instead, we explicitly state the uncertainties in TOC values beforehand (L229-235).
 - Blair and Aller 2013 is a good reference to consider for OC diagenesis
 - L235 - ref. 41
 - References. 19,20,21 are these available?
 - These references are no longer included. Ref. 19 was replaced with ref. 58 and ref. 21 has been replaced by ref. 33. The author of ref. 20 is now a co-author and the dissertation is available on request.

Reviewer #2 (Remarks to the Author):

Tectonically-driven changes in tidal dynamics drive mangrove carbon sequestration during the Oligo-Miocene

In my opinion this paper has the potential to become suitable for Nature communications, as it brings a very interesting message forward. However, in my opinion, it is not suitable in its present form, for the following two main reasons:

- the paper is not written so that it can be easily read by a broad audience. If you are not into Palaeogeography, you basically need to go to the web to be able to read the text Same for location names; makes you miss a proper map. In my opinion, every paper in Nature communications should be easily readable by people outside the research field. But this is a trivial point, as it can be easily improved by some rewriting
 - We have adopted this suggestion: the paper has been extensively rewritten, lengthened and new summary figures are included e.g. Fig. 5, 7e and 8.

• The main problem that I have with the paper is that it does not get clear to me to which extent having an increased tidal range / tidal dynamics is essential. Figure 1a, 1b and 1c, shows that part of the land gets flooded over time. This suggest that mangroves would gradually shift inward over time, and sequester carbon in the process. So the question I would like to see explicitly answered is: to what extent might this process alone explain the carbon sequestration by mangroves, regardless wether there was a changes in tidal range or not? How much does an altered tidal range add on top of this? The paper states " The elevated tidal range and tidal bed shear stress optimized the development of 18 a vast mangrove biome and associated carbon-rich sediments bordering the SCS. Over 19 geological time scales (106 years) ", suggesting the change in tidal range to be of crucial importance. But as far as I understand, there is not comparison to the situation not accounting for a change in tidal range.

- We explicitly refer to the possibility that OC sequestration increased independent of tidal range etc. due to an overall rise in sea level in the western SCS (L324-328). However, we have now: (1) expanded discussion on the complexities of preserving sediment during short-term RSL cycles (L328-337and L362-387; and Fig. 1); (2) expanded the thought-process regarding the impact of elevated tidal range and tidal bed shear stress on the extent and productivity of the mangrove biome (L305-323 and Fig. 8). The change in tidal range and tidal strength have several important possible effects that we strongly suggest encourage increased mangrove development and OC burial beyond that expected due to a change in long-term accommodation.

Please be aware that I am not a specialist in this topic, and if I here raise a question that is obviously not relevant, please ignore this comment. But as is, it was not clear to me.

Hope this feedback allows the authors to submit a revised manuscript, as the topic is very interesting

Reviewer #3 (Remarks to the Author):

This is an interesting paper that links Oligocene-Miocene paleoenvironmental and tidal reconstructions around the South China Sea to the existence of a large productive mangrove biome and addresses impacts to the global carbon cycle. The tidal modeling component is emphasized in the paper. I would have liked to have seen more emphasis placed on supporting the paleoenvironmental reconstructions and the distribution of mangroves through time.

- We have expanded the section on palaeoenvironmental reconstructions (L74-117) and included Figs 2-3.
- The preserved record of mangrove influenced sedimentation has been re-reviewed (L141-210, Figs. 6-7).
- The controls on and possible changes in distribution of mangroves through time are discussed more extensively in L305-323.

Most of the results presented appear to be based on gross estimates and assumptions. For example, much of the text focuses on the Baram Delta Province. The authors appear to

apply information from that system to four other large deltas in the SCS to scale carbon sequestration numbers to the larger basin.

- We now present a newly compiled regional archive of Oligo–Miocene sedimentary and stratigraphic data from twelve SCS basins (Fig. 6), in addition to the analysis of Early–Middle Miocene, tide- and mangrove-influenced sedimentary rocks in northwest Borneo. We compare the regional sedimentary data archive directly to our tidal modelling results (Fig. 6).
- We assert that applying information from the BDP to the wider SCS region is justified because deposition in all these basins occurred in similar tropical, wave- and tide-influenced, coastal-deltaic to deep marine systems, with abundant evidence for mangrove vegetation (Fig. 6), as manifested in the modern Baram, Mekong, Pearl River, Rajang and Red River Deltas (L226-229).
- We have revised the text on the methods and assumptions regarding our OC burial estimates (L211-235), including justifying the use of estimates made in the BDP to the wider SCS region (L223-229). See also Methods L473-490.

It is not clear why the TOC values for the Bengal Fan should be applied to the SCS.

- We no longer apply TOC values from Bengal Fan to SCS. We justify the assumed TOC values in L232-235.

Overall I found that the main point of the study, that the mangrove biome in the SCS contributed significantly to the global carbon cycle on geological timescales, could be true, but is not supported well with the information presented here.

- We have added a lot more information on: (1) palaeoenvironmental interpretations, particularly mangrove sedimentation (L141-210, Fig. 6); (2) the possible impact of changing tidal regime on mangroves (L305-323); and (3) preservation of mangrove sediment and OC burial (L324-387).

Line 12: It is not only tidal range that determines the areal extent of the mangrove habitat. The substrate depth is just as important. In addition, tidal currents are not the primary determinant on the rate of mangrove sediment deposition because the rate of sea-level rise is extremely important in this regard.

- We have extensively modified our discussion on the control of tides on mangrove distribution, productivity, sediment accretion and OC burial (L305-323 and Fig. 8). Further discussion has been made on the effect of relative sea level rise on accommodation space creation and mangrove OC preservation (e.g. L360-387).

It would be nice if the authors would include the geologic record of atmospheric CO₂ concentrations so the reader can put the stated "it could have contributed to a decrease in atmospheric CO₂ concentrations of 900 ppm since the mid-Miocene" in perspective to other fluctuations in CO₂. Over what period of time, exactly, do the authors think the reduction occurred over? This is important information to include in the first paragraph. Mid-Miocene to present is very broad.

- We no longer make this statement regarding decreases in CO₂ ppm. We instead suggest that burial of mangrove OC carbon in the SCS was a contributory factor to the overall decrease in CO₂ ppm during the Oligo-Miocene (L272-276). There remain uncertainties in the estimates (discussed in L226-235, L247-250 and L253-260). The inclusion of Table 1 is aimed at making it clearer the various estimates of OC burial

for different regions and time periods.

The link between the tidal modeling and the impact on the mangrove biome is not supported well with hard data. It is not clear to this reader exactly how increased tidal energy promotes increased preservation of mangroves (Line 74).

- We have extensively revised the explanation on how we suggest increased tidal range and energy promotes (1) increased mangrove development (L305-323), and (2) how mangrove sediments are preserved (L197-205). We make the link between tides being stronger/higher, coastal morphology, mangrove development (e.g. Fig. 8) and the environments in which tide dominated stratigraphy (including mangrove sediments) is preserved (L141-210 and L333-352), and compare this to a major modern system with abundant mangroves (the Ganges-Brahmaputra; L352-359).

Extended data figure 1. This is an important figure, but difficult to digest, because environmental classifications are vague and colors used are not distinctive. I cannot tell the difference between Lacustrine and Coastal on the maps. I cannot find the intertidal color on the maps. The continental slope is only defined as a water depth, which is incorrect because it is a physiographic feature. There are areas on the map with that color that are not part of the slope (but again difficult to tell with the similarity in colors). What does "coastal" mean? It seems like coastal should be indicated everywhere there is a coast, but that is not the case.

- We have converted the palaeoenvironmental maps to the palaeobathymetric interpretations and included the ancient reconstructions, together with modern, in Fig. 2.
- "Coastal" has been changed to refer to "Delta plain" (caption of Supplementary Fig. 1).

On line 253, "a -50 m sea level transgression" is odd phrasing to me. -50 m implies sea-level fall. Essentially, the palaeobathymetric uncertainty is the entire depth of the shallow shelf? That seems high when the focus of the paper is on Mangroves that live in shallow water. I might be missing something here.

- It is extremely difficult to generalize the palaeobathymetric uncertainty across the whole area and from basin to basin: it varies for each basin, it depends on the abundance of available data and on the rigor and quality of previous interpretations. We have modelled all time-averaged base-case palaeogeographic reconstructions to the sea level highstand for that timeslice. However, we also produced reconstructions for a sea-level lowstand for which sea level was decreased by 50 m. The original statement - "a -50 m sea level transgression" - was badly phrased and is no longer used (L412-415). 50 m is the average sea level variability during the Oligo-Miocene (L411-412), therefore there is an estimated 50 m water depth variability in the shoreline position during this period.

I don't understand what >2000 is referring to on line 256. Is this >2000 meters of mangrove-bearing, Miocene coastal-deltaic strata or >2000 samples, or something else. What is the spatial distribution of the samples? How did you interpolate between observations?

- This was a typing error and should have been > 2000 m of ...strata (L428-433). The studied outcrops have relative age dates and can be correlated into different

formations of the same approximate ages (L78-182 and L428-433)

Line 161. Is that the global sedimentary OC reservoir that is being referred to? I'm a bit lost here. Why is the TOC percentage from the Bengal Fan being applied to the SCS on lines 162 and 163?

- This section has been rewritten, restructured and expanded (L211-284). We now explicitly refer to the global sedimentary OC reservoir during the Neogene (L279-282; ref. 44) and no longer refer to the Bengal Fan.

Reviewer #4 (Remarks to the Author):

Dear authors,

The manuscript "Tectonically-driven changes in tidal dynamics drive mangrove carbon sequestration during the Oligo-Miocene" provide insights into the importance of carbon storage by mangroves over geological timescales, from the Oligocene to present. The topic is exciting and novel, and the approach seems sound although my lack of expertise on modelling precludes a comprehensive judgement of this manuscript. My main concern relies on the number of assumptions required to reach the conclusions raised by the authors, such as the uncertainties of the models, and the estimates of volume of organic deposits and hydrocarbons.

- **Uncertainty in model:** We acknowledge that there is uncertainty in the palaeotidal modelling, because of the inherent uncertainty in constructing ancient palaeogeographic maps. The maps constructed use a fully-global plate tectonic model (obtained from Getech), which is underpinned by extensive published and unpublished data, has undergone extensive validation against previous interpretations and data (e.g. Mazur et al. 2012), and is constantly evolving to include new data and interpretations. The tidal model has been extensively validated for the present-day global tides and in several sub-regions (e.g. refs. 26 and 45)
- **Uncertainty in the volume of organic deposits and hydrocarbons:** We acknowledge that these estimates are, at this stage, merely a guide to the possible amount of OC burial. The hydrocarbon estimates are based on the most recently published and comprehensive hydrocarbon in-place or reserve estimates (e.g. refs. 17, 23 and 39). We have fully acknowledged a range of uncertainties in the range of estimates we have made e.g. L226-235, L247-250 and L253-260). The inclusion of Table 1 is aimed at making it clearer the various estimates of OC burial for different regions and time periods.

Major comments:

1. The authors discussed the importance of tidal range and shear bed stress in driving OC storage and mangrove area, but 'little' or none mention to how changes in sea level rise may impacted OC storage. Thus, coastal geomorphology (slope), which vary across the SCS, may have a significant effect on the area of mangrove. Indeed, the manuscript only focusses in OC storage, but the e.g drop in sea level (over glacial/inter-glacial periods) would expose the OC-rich soil mangrove to oxygen, increasing CO₂ emissions (ppm) and ultimately compensating/enhancing CO₂ due to changes in mangrove area or tide bed shear stress.

Please discuss how all above may influenced OC storage (balance) over geological scales.

- The possible effects of relative sea level changes (both decrease and increase) on (1) the accommodation space for mangrove development and accumulation, and (2) preservation of mangrove material on geological timescales, is now extensively discussed in L324-337 and L360-387). We have included more detail in our depositional environment interpretation of mangrove material in L178-205 and L344-364. We have also added further detail to our interpretation of the effects of increases in tidal range and bed shear stress on mangrove systems (L338-359, Fig. 8).

2. L22-23. Estimates of atmospheric CO₂ concentration (ppm) over the Miocene point to values around 1000 ppm, but I don't follow how the TOC sequestration by mangrove could led to a decrease in atmospheric CO₂ concentrations of c. 900 ppm. Please explain how you reach this conclusion and if it does make sense in a broader context.

- We acknowledge that the original wording of this statement was misleading. We no longer make this statement and instead suggest that burial of mangrove OC carbon in the SCS was a contributory factor to the overall decrease in CO₂ ppm during the Oligo-Miocene (L272-276).

Minor comments:

L14: 170 g OC m⁻² yr⁻¹ instead of 170 gC m⁻² yr⁻¹

- L28-29: corrected

L16-18: this hypothesis is not supported in the sections below. Please clarify

- Now stated on L124-125 but corrected to 'Late Oligocene–Early Miocene'. Tidal simulations were made on global palaeogeographic interpretations (Methods L449-451), therefore, we make the observation in the model results. The plots in Fig. 2 and Supplementary Figs 2-8 illustrate that tides in the SCS were >8 m (and often > 10 m).

L18. Elevated tidal range will increase the areal extent of mangrove area, but elevated tidal bed shear stress may cause sediment erosion rather than accumulation of carbon-rich sediments. seems contradictory to me. Please rephrase or reconsider this hypothesis.

- This statement is no longer included in this form. The potential erosive effects of increased tidal bed shear stress are acknowledged in L314-315 and the occurrence of modern mangroves along high energy coastlines, and their ability to regenerate after destruction, are mentioned and discussed in L318-323.

L63: "...due to northward migration..." instead of "...due northward migration of..."

- L296: corrected

L79-82: Increase sediment exchange/tidal bed shear stress would probably resulted in OC export and remineralization, overall decreasing OC accumulation. Please clarify.

- Clarified on L309-314.

L120: add reference to the meaning of BDP here

- L180: corrected

L145: replace TOC for OC.

➤ L253: rephrased

Reviewers' Comments:

Reviewer #1:

Remarks to the Author:

This paper is greatly improved from when I last reviewed it. Well done to the authors for improving the writing and making it easier for the non-geologist to understand. It is an interesting study.

Some further suggestions for improvements of this version are:

1. line 31 atmospheric carbon sequestration (1110 g OC m⁻² yr⁻¹)³. I'm guessing this is Net Primary Production which is not carbon sequestration. I would omit this clause – or say mangrove forests have high rate of primary productivity.
2. Line 37 – is there a simple, more straight forward definition of stratigraphic architecture? I'm guessing what stratigraphic architecture is.
3. Line 44 – what is meant by rates of accommodation? Can you be clearer what this means. We usually refer to accommodation space, but not a rate.
4. Line 70 "during the Oligocene–Present-day." Replace with words - During the Oligocene to the Present
5. Line 70 tectonophysiographic - this is jargon.
6. Line 148 "In this setting...." Which setting are you referring to? Please clarify.
7. Line 149 – what is the "calibre" of sediment? Is this jargon? Does it mean grain size distribution of the sediment?
8. Line 160 – what is "dune scale" and how is it relevant here? This is confusing –dunes are sand dunes and this paper is about coastal wetlands, which are not dunes.
9. Line 161 – 163 – this sentence is tacked on the end here and there is no indication of what it means or what the reader is supposed to understand from this fact. It is an inappropriate sentence for the closing sentence of a paragraph.
10. Line 168 – Dune scale again
11. Line 208/216 – BDP – was this defined earlier in the text?
12. L221 – missing a full-stop
13. Line 420 and throughout - you need a space between units mm yr⁻¹
14. Line 244 – what is tcf?
15. Line 244 – "initially in place" does this mean before extraction by the oil and gas industry? Please clarify
16. Line 246 Is 0.9 ppm supposed to be a rate, as is expressed in L252-253?
17. Line 254 – which sample set was biased – please clarify
18. L332 tectono-stratigraphic styles – what does this mean? Is this jargon?
19. L333 – an incomplete sentence – more than what? Regressive phases?
20. L338 – this paragraph is really difficult to follow. Please start with a topic sentence that is intelligible to the non-geologist reader. Keep to the point.
21. Line 340 – what are tide dominated units?
22. Line 374 "SCS basins have undergone >5 km subsidence and sedimentation since the Oligocene" Is this sentence suggesting subsidence and sedimentation have similar magnitude? Can you provide a value for sedimentation as you have for subsidence?
23. Line 389. I found this a very unsatisfactory last sentence "During the Neogene in the SCS, tectonic-driven changes in basin configuration led to a decrease in tidal influence and hence the size and 390 productivity of the mangrove biome; the scale of these changes could be a significant 391 component of the global carbon cycle on geological timescales." This says the main point of the paper is that the role of mangroves in the carbon cycle has been declining over time. Yet the abstract has the message that mangrove forests have had a positive influence on global C sequestration. Can you please harmonize?
24. In Fig 1 there is a -6 that should not be there (above the arrows)
25. Figure 3B Need more information to say the island of Borneo is outlined in black and what the contours are - provide more information to make this map easier to understand.
26. Fig 5 - what are the dashed lines? More information is needed

27. Fig 6 and 7 - these are nice, but there are a lot of figures. Can these go to the supplementary. This is an editorial decision.

Reviewer #3:

Remarks to the Author:

This is an interesting paper, and I believe that my comments have been addressed appropriately with the revision. The study should be of interest to sedimentologists, stratigraphers and paleoclimatologists, especially those who work in the Neogene. The authors back off many of the most interesting assertions presented in the first version of the manuscript because of uncertainties. While that is definitely the correct approach, this new version of the paper is less impactful. The most important finding is that the decrease in the size and productivity of the mangrove biome during the Neogene in the SCS could be a significant component of the global carbon cycle on geological timescales; which is basically the last sentence of the manuscript.

I would avoid the wording used on line 124. "The SCS experienced the highest tides (>8 m tidal range) on Earth during the Late Oligocene-Early Miocene." What is the evidence to support this statement? I don't think the modeling results presented are global in scale. Currently, there are tidal ranges on Earth that exceed 8 m. One well-known example is the Bay of Fundy.

Reviewer #4:

Remarks to the Author:

Dear authors,

Thank you for addressing the main concerns raised by the reviewers in this new version of the manuscript. This new version acknowledges the limitations/assumptions linked to palaeoreconstructions based on indirect approaches such as modelling, and clarifies some of the 'risky' hypothesis raised in the first version. In my opinion this manuscript can be accepted for publication in the present format.

I would like to encourage the authors to collect and study soil cores in the area to test the hypothesis raised in this manuscript in the future.

Numbered = Reviewer comment

Bullet = Response

NB. Line numbers are for Full Markup version

REVIEWERS' COMMENTS:

Reviewer #1 (Remarks to the Author):

This paper is greatly improved from when I last reviewed it. Well done to the authors for improving the writing and making it easier for the non-geologist to understand. It is an interesting study.

Some further suggestions for improvements of this version are:

1. line 31 atmospheric carbon sequestration (1110 g OC m⁻² yr⁻¹)³. I'm guessing this is Net Primary Production which is not carbon sequestration. I would omit this clause – or say mangrove forests have high rate of primary productivity.
 - Now omitted from its original position and placed in L30 and references as 'primary productivity' as advised.
2. Line 37 – is there a simple, more straight forward definition of stratigraphic architecture? I'm guessing what stratigraphic architecture is.
 - L39: Changed to 'character of sedimentary layering (stratigraphic architecture)'
3. Line 44 – what is meant by rates of accommodation? Can you be clearer what this means. We usually refer to accommodation space, but not a rate.
 - L46: Now changed to 'rate of accommodation space creation' – accommodation space can change through time at various rates (please refer to Catuneanu et al. (2009) for a review of the widely applied geological concept of sequence stratigraphy).
4. Line 70 "during the Oligocene–Present-day." Replace with words - During the Oligocene to the Present
 - Replaced here and throughout
5. Line 70 tectonophysiographic - this is jargon.
 - This term has now been removed in the revised last paragraph of the introduction.
6. Line 148 "In this setting...." Which setting are you referring to? Please clarify.
 - L155-156: Amended to 'In shoreline–shelf depositional systems'
7. Line 149 – what is the "calibre" of sediment? Is this jargon? Does it mean grain size distribution of the sediment?
 - L156: Changed to 'grain size distribution'

8. Line 160 – what is “dune scale” and how is it relevant here? This is confusing dunes are sand dunes and this paper is about coastal wetlands, which are not dunes.

- L169: Changed to ‘decimetre- to metre-scale’ here and throughout. FYI ‘dune-scale’ is a term used to describe type and size of sedimentary bedforms e.g. Baas et al. (2016).

9. Line 161 – 163 – this sentence is tacked on the end here and there is no indication of what it means or what the reader is supposed to understand from this fact. It is an inappropriate sentence for the closing sentence of a paragraph.

- L161-163 now been moved to L182-185, and has been amended to include a comparison the model results (Fig. 4a, d). This adds additional support to the assertion (L183-187) that the palaeo-Gulf of Thailand was tidally influenced and facilitated mangrove colonization throughout the Mio–Pliocene.

10. Line 168 – Dune scale again

- L181: Changed to ‘decimetre- to metre-scale’ (point 8).

11. Line 208/216 – BDP – was this defined earlier in the text?

- Defined on L192.

12. L221 – missing a full-stop

- Inserted L231.

13. Line 220 and throughout - you need a space between units mm yr⁻¹

- Applied throughout.

14. Line 244 – what is tcf?

- Now trillion cubic feet (tcf) (L254).

15. Line 244 – “initially in place” does this mean before extraction by the oil and gas industry? Please clarify

- L255: Clarified to ‘gas initially in place (before production)’.

16. Line 246 Is 0.9 ppm supposed to be a rate, as is expressed in L252-253?

- L257: Now amended to the rate calculation (0.1 ppm CO₂ per Myr) for consistency.

17. Line 254 – which sample set was biased – please clarify

- L265: Amended to ‘western BDP sample set’ (L258).

18. L332 tectono-stratigraphic styles – what does this mean? Is this jargon?

- L341-342: Changed to ‘stratigraphic patterns reflecting different tectonic and basin-fill histories’. Tectonostratigraphy is a valid geological term that describes the stratigraphic patterns formed by the effects of active tectonics (<https://en.oxforddictionaries.com/definition/tectonostratigraphic>).

19. L333 – an incomplete sentence – more than what? Regressive phases?
- L343-345: Changed to include ‘than regressive phases’ and make point clearer that tide-influenced coastlines are more common (not that coastlines in general are more tidally influenced)
20. L338 – this paragraph is really difficult to follow. Please start with a topic sentence that is intelligible to the non-geologist reader. Keep to the point.
- L350-376: First sentence (topic sentence) heavily modified with simplified language (e.g. ‘stratigraphic units’ instead of ‘facies successions’) and explanation of key terms (e.g. allogenic and autogenic)
 - Paragraph extensively modified to improve flow and linkage of allogenic vs. autogenic vs. modern Ganges- Brahmaputra delta
 - Inserted new sentence to explain why we compare to the Ganges- Brahmaputra delta (e.g. L366-376)
21. Line 340 – what are tide dominated units?
- L350: Amended to ‘tide- and wave-dominated stratigraphic units’
22. Line 374 “SCS basins have undergone >5 km subsidence and sedimentation since the Oligocene” Is this sentence suggesting subsidence and sedimentation have similar magnitude? Can you provide a value for sedimentation as you have for subsidence?
- L392: Deleted ‘and sedimentation’ as Fig. 3. refers to subsidence.
23. Line 389. I found this a very unsatisfactory last sentence “During the Neogene in the SCS, tectonic driven changes in basin configuration led to a decrease in tidal influence and hence the size and productivity of the mangrove biome; the scale of these changes could be a significant component of the global carbon cycle on geological timescales.” This says the main point of the paper is that the role of mangroves in the carbon cycle has been declining over time. Yet the abstract has the message that mangrove forests have had a positive influence on global C sequestration. Can you please harmonize?
- L405-411: The last sentence has been rewritten to focus on the optimized conditions for mangrove development and preservation in the Oligo-Miocene and the positive influence (i.e. increased) mangrove OC burial had on the global carbon cycle. L398-404.
24. In Fig 1 there is a -6 that should not be there (above the arrows)
- Corrected. Also added colour (editor comment).
25. Figure 3B Need more information to say the island of Borneo is outlined in black and what the contours are - provide more information to make this map easier to understand.
- L557-559: Added Borneo label and modified caption to highlight coastline and state what contours represent
26. Fig 5 - what are the dashed lines? More information is needed
- L574: Dashed lines show alternative trend for 6 Ma model results with submerged IBM arc. This has been added to caption

27. Fig 6 and 7 - these are nice, but there are a lot of figures. Can these go to the supplementary. This is an editorial decision.

- Figures 6 and 7 are very important figures including significant amounts of data supporting model results (Fig. 6), the sedimentary records of SCS basins (Figs 6 and 7) and examples of mangrove facies (Fig. 7). We strongly urge the figures to be kept within the main text.

Reviewer #3 (Remarks to the Author):

This is an interesting paper, and I believe that my comments have been addressed appropriately with the revision. The study should be of interest to sedimentologists, stratigraphers and paleoclimatologists, especially those who work in the Neogene. The authors back off many of the most interesting assertions presented in the first version of the manuscript because of uncertainties. While that is definitely the correct approach, this new version of the paper is less impactful. The most important finding is that the decrease in the size and productivity of the mangrove biome during the Neogene in the SCS could be a significant component of the global carbon cycle on geological timescales; which is basically the last sentence of the manuscript.

I would avoid the wording used on line 124. "The SCS experienced the highest tides (>8 m tidal range) on Earth during the Late Oligocene-Early Miocene." What is the evidence to support this statement? I don't think the modeling results presented are global in scale. Currently, there are tidal ranges on Earth that exceed 8 m. One well-known example is the Bay of Fundy.

- All tidal models were global (as stated on L468), consequently, we can demonstrate (upon request) that the modelled tidal ranges in the Late Oligocene-Early Miocene were the highest on Earth.

Reviewer #4 (Remarks to the Author):

Dear authors,

Thank you for addressing the main concerns raised by the reviewers in this new version of the manuscript.

This new version acknowledges the limitations/assumptions linked to palaeoreconstructions based on

indirect approaches such as modelling, and clarifies some of the 'risky' hypothesis raised in the first version.

In my opinion this manuscript can be accepted for publication in the present format.

I would like to encourage the authors to collect and study soil cores in the area to test the hypothesis raised

in this manuscript in the future.

- A project proposal has been submitted to collect and study peat and sediment cores from the Holocene to Present Mekong River delta.

References

- BAAS, J. H., BEST, J. L. & PEAKALL, J. 2016. Predicting bedforms and primary current stratification in cohesive mixtures of mud and sand. *Journal of the Geological Society*, 173, 12-45.
- CATUNEANU, O., ABREU, V., BHATTACHARYA, J. P., BLUM, M. D., DALRYMPLE, R. W., ERIKSSON, P. G., FIELDING, C. R., FISHER, W. L., GALLOWAY, W. E., GIBLING, M. R., GILES, K. A., HOLBROOK, J. M., JORDAN, R., KENDALL, C. G. S. C., MACURDA, B., MARTINSEN, O. J., MIALL, A. D., NEAL, J. E., NUMMEDAL, D., POMAR, L., POSAMENTIER, H. W., PRATT, B. R., SARG, J. F., SHANLEY, K. W., STEEL, R. J., STRASSER, A., TUCKER, M. E. & WINKER, C. 2009. Towards the standardization of sequence stratigraphy. *Earth-Science Reviews*, 92, 1-33.